# Switching of band inversion and topological surface states by charge density wave

N. Mitsuishi [1,2], Y. Sugita[1], M. S. Bahramy [1,2,3], M. Kamitani[1,2], T. Sonobe [1,2], M. Sakano[1,2], T. Shimojima[3], H. Takahashi[4], H. Sakai [5], K. Horiba[6], H. Kumigashira[6], K. Taguchi[7], K. Miyamoto[7], T. Okuda [7], S. Ishiwata [4], Y. Motome [1] & K. Ishizaka [1,2,3] ✉

Topologically nontrivial materials host protected edge states associated with the bulk band inversion through the bulk-edge correspondence. Manipulating such edge states is highly desired for developing new functions and devices practically using their dissipation-less nature and spin-momentum locking. Here we introduce a transition-metal dichalcogenide $VTe_2$, that hosts a charge density wave (CDW) coupled with the band inversion involving V3$d$ and Te5$p$ orbitals. Spin- and angle-resolved photoemission spectroscopy with first-principles calculations reveal the huge anisotropic modification of the bulk electronic structure by the CDW formation, accompanying the selective disappearance of Dirac-type spin-polarized topological surface states that exist in the normal state. Thorough three dimensional investigation of bulk states indicates that the corresponding band inversion at the Brillouin zone boundary dissolves upon the CDW formation, by transforming into anomalous flat bands. Our finding provides a new insight to the topological manipulation of matters by utilizing CDWs' flexible characters to external stimuli.

[1] Department of Applied Physics, The University of Tokyo, Tokyo 113-8656, Japan. [2] Quantum-Phase Electronics Center (QPEC), The University of Tokyo, Wako 113-8656, Japan. [3] RIKEN Center for Emergent Matter Science (CEMS), Wako 351-0198, Japan. [4] Division of Materials Physics, Graduate School of Engineering Science, Osaka University, Toyonaka, Osaka 560-8531, Japan. [5] Department of Physics, Osaka University, Toyonaka, Osaka 560-0043, Japan. [6] Condensed Matter Research Center and Photon Factory, Institute of Materials Structure Science, High Energy Accelerator Research Organization (KEK), Tsukuba 305-0801, Japan. [7] Hiroshima Synchrotron Radiation Center (HSRC), Hiroshima University, 2-313 Kagamiyama, Higashi-Hiroshima 739-0046, Japan. ✉email: ishizaka@ap.t.u-tokyo.ac.jp

Since the discovery of topological insulators, a wide variety of topological phases have been intensively developed and established in realistic materials[1,2]. The upcoming target is to explore the guiding principles for the manipulation of these topological states. A key parameter to characterize the topological nature of materials is the band inversion realized by the crossing and anti-crossing of energy bands with opposite parities. In bulk materials, the control of band inversion has been mostly done through the tuning of spin-orbit coupling (SOC) by element substitution[3,4], with some exceptions related to the topological crystalline insulators[5]. In the present work, we focus on charge density wave (CDW), i.e., the spontaneous modulation of charge density and lattice that modifies the periodicity and symmetry of the host crystal. In $TaS_2$, for example, the Star-of-David type CDW superstructure induces the effective narrowing of valence bands thus causing the Mott transition[6–8]. From this viewpoint, CDW could also modify the band structures that involve the band inversion, and induce topological change. Moreover, it is worth noting that CDW can be flexibly controlled by external stimuli.

The transition metal dichalcogenides (TMDCs) are a well-known family of layered materials that host a variety of CDWs reflecting their quasi-two dimensionality[9]. Manipulations of CDW states have been intensively investigated and realized, especially in the aforementioned archetypal material $TaS_2$, by various stimuli, such as pressure[7], electric field[10], and optical pulse[11,12]. The feasibility of CDW thus paves the way toward triggering the exotic phase transitions and generating new functions. More recently, there has been increasing interest in SOC effect in TMDCs[13–18]. Especially in the tellurides, (Mo,W)$Te_2$[14–16] and (Ta,Nb)$IrTe_4$[17,18] are reported to be topological Weyl semimetals. Here, the peculiar quasi-one-dimensional chain-like structures inherent to tellurides, as well as their stronger SOC compared with selenides and sulfides, are essential in realizing the topologically non-trivial states. In this stream, we focus on the telluride material $VTe_2$, to investigate the interplay of band topology and CDW instability.

$VTe_2$ has $CdI_2$ structure in the high-temperature normal $1T$ phase, consisting of trigonal layers formed by edge-sharing $VTe_6$ octahedra (Fig. 1a). With cooling, it undergoes a phase transition to the CDW phase at around 475 K, appearing as a jump in the temperature-dependent resistivity[19]. The resulting CDW state exhibits a $(3 \times 1 \times 3)$ superstructure characterized by double zig-zag chains of vanadium atoms (Fig. 1d; hereafter we refer it as $1T''$ phase, see also Supplementary Note 1)[19–21]. This superstructure commonly appears in group-V transition metal ditellurides $MTe_2$ ($M$ = V, Nb, Ta)[20,22]. Its relation with CDW has been discussed to originate from the peculiar chemical σ-bonding among $t_{2g}$ $d$-orbitals connecting three adjacent $M$ sites, in, for example, other TMDCs including $TaS_2$[23,24] and purple bronze $AMo_6O_{17}$ ($A$ = Na, K)[25–27]. Previous band calculation on $NbTe_2$, on the other hand, implies the importance of Fermi surface nesting and electron-phonon coupling as the origin of CDW with $1T''$ lattice distortion[28]. However, the modifications of electronic structure by this CDW transition have not been systematically investigated so far. It is worth noting that the contraction of metal-metal bond length ($\Delta a_{max}/a$) via the CDW transition in $VTe_2$ is fairly large (~9.1%)[20] and comparable with that in $TaS_2$ (~7.0%)[29]. We also note that the CDW in $VTe_2$ can be optically manipulated, as reported by recent time-resolved diffraction studies[30,31].

In this Article, we investigate the electronic structures of $VTe_2$, by employing spin- and angle-resolved photoemission spectroscopy (ARPES) and first-principles calculations. We focus on the modifications of the bulk bands through the CDW formation, and its relevance to the Dirac surface states stemming from the topological band inversion. Experimentally it is not easy to perform highly precise ARPES measurement on the normal $1T$ phase

of pristine $VTe_2$ (>475 K), since such a high temperature may cause the sample degradation. Therefore we start from $TiTe_2$ showing a simple $1T$ structure down to the lowest temperature[32], and develop the single crystalline $V_{1−x}Ti_xTe_2$ to access both $1T$ and $1T''$ phases at appropriate temperatures. Figure 1f displays the schematic electronic phase diagram of $V_xTi_{1−x}Te_2$, based on the temperature-dependent ARPES measurements. With increasing Ti, the CDW phase transition becomes gradually suppressed to the lower temperature region (see Supplementary Note 2). For investigating both the normal and CDW phases within a single sample, we synthesized the minimally Ti-doped single crystals ($0.10 \leq x \leq 0.13$) showing the transition just below room temperature (280–250 K). To discuss the electronic structures in momentum space, we introduce the following notation of the Brillouin zones (BZ). In $1T$, the 2-dimensional (3-dimensional) BZ is represented by the hexagonal plane (prism) reflecting the trigonal symmetry (Fig. 1b, c). In $1T''$, the BZ changes into a smaller one of lower symmetry as indicated in Fig. 1e. In this paper, we use the $1T$ BZ notation to present the band structures in $1T$ and $1T''$ in a common fashion. To account for the in-plane anisotropy of vanadium chains that is essential in $1T''$, we define $\bar{K}_1$, $\bar{K}_2$ and $\bar{M}_1$, $\bar{M}_2$ as depicted in Fig. 1e (see Supplementary Note 1 for the details of BZ).

## Results

**Overviewing the electronic structures and modification by CDW.** Let us start by briefly overviewing the anisotropic modification of electronic structures from $1T$ to $1T''$ by presenting the ARPES data successfully focused on a CDW single-domain region. Figure 2a shows the ARPES image for the normal-state $1T$-$V_{0.87}Ti_{0.13}Te_2$ taken at 300 K with a He-discharge lamp (photon energy $h\nu = 21.2$ eV). We find V-shaped band dispersions along $\bar{K} - \bar{M} - \bar{K}$ that clearly cross the Fermi level ($E_F$). Looking at the higher binding energy ($E_B$) region, these V-shaped bands are connected to the Dirac-cone-like bands reminiscent of surface states in topological insulators, with the band crossing (Dirac points) at $\bar{M}$. The topological character of these Dirac bands will be discussed later. On the other hand, Fig. 2b displays the ARPES results on a single-domain CDW state in $1T''$-$VTe_2$ (200 K, synchrotron light $h\nu = 90$ eV). Because of the zigzag type CDW formation, the system now loses the threefold rotational symmetry, and the three equivalent $\bar{M}$ points in $1T$ turn into one $\bar{M}_1$ and two $\bar{M}_2$. Here, the V-shaped band and Dirac-like bands remain at $\bar{M}_1$, whereas at $\bar{M}_2$ side the unusual flat band is observed and the Dirac-like state is vanished. Thus, the $1T$-$1T''$ CDW transition induces the huge directional change of electronic structure accompanying the selective disappearance of Dirac-like states. In the following, we discuss these band structures in detail, by comparing with band calculations.

Figure 3a, b respectively displays the bulk band calculation for the normal-state $V_{0.87}Ti_{0.13}Te_2$ along high symmetry lines (Γ-K-M-Γ) and the Fermi surface at $k_z = 0$ (see Supplementary Note 3). They are characterized by the circular and triangular hole Fermi surfaces respectively around Γ and K. The ARPES results on the normal-state $V_{0.90}Ti_{0.10}Te_2$ (350 K, marked as "#1" in Fig. 1f) are shown in Fig. 3c, d. Here a He-discharge lamp ($h\nu = 21.2$ eV) is used as the light source. The black broken curves in Fig. 3c are the guides for the bands plotted by tracking the peaks of the energy/momentum distribution curves (EDCs/MDCs) (see Supplementary Note 4). We find hole-like bands centered at around $\bar{Γ}$ and $\bar{K}$ points, in a qualitative agreement with the calculation in Fig. 3a (We note that the tiny electron pocket at Γ is not clearly detected, probably reflecting the finite energy mismatch of the bands compared with the calculation.). In the EDCs along the $\bar{M} - \bar{K}$ line (Fig. 3d), we can clearly trace a dispersive band that crosses $E_F$.

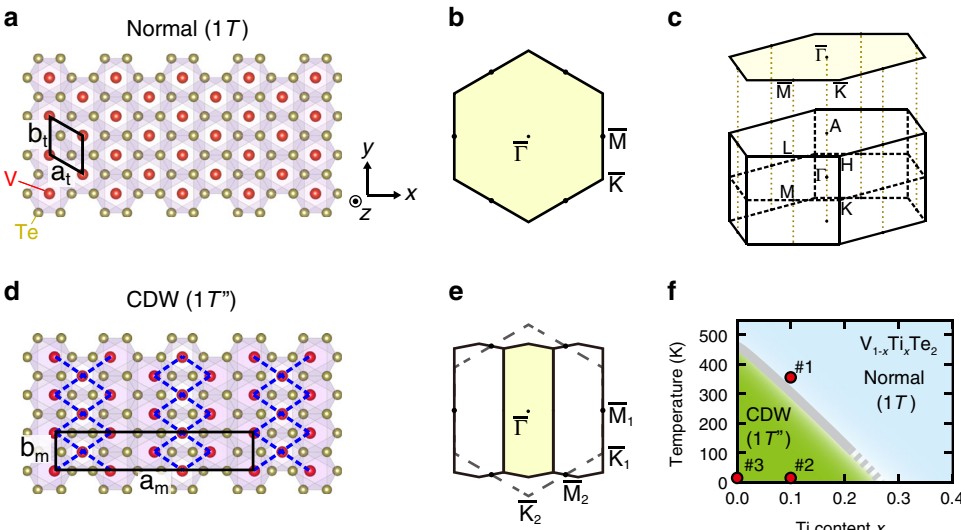

**Fig. 1 Normal (1$T$) and CDW (1$T''$) phases in VTe$_2$ system. a** Top view of VTe$_2$ layer in the high-temperature normal 1$T$ phase. The conventional unit cell is indicated by the black lines. **b** (0 0 1) surface BZ of 1$T$-VTe$_2$. **c** Bulk BZ of 1$T$-VTe$_2$. **d** Top view of VTe$_2$ layer in the low-temperature CDW 1$T''$ phase. The blue broken lines highlight the shortest V–V bonds forming the double zigzag chains. **e** (0 0 1) surface BZ considering the CDW in 1$T''$-VTe$_2$, superimposed on that in the 1$T$ phase (dashed hexagon). **f** Schematic electronic phase diagram for V$_{1-x}$Ti$_x$Te$_2$, based on the temperature-dependent ARPES measurements.

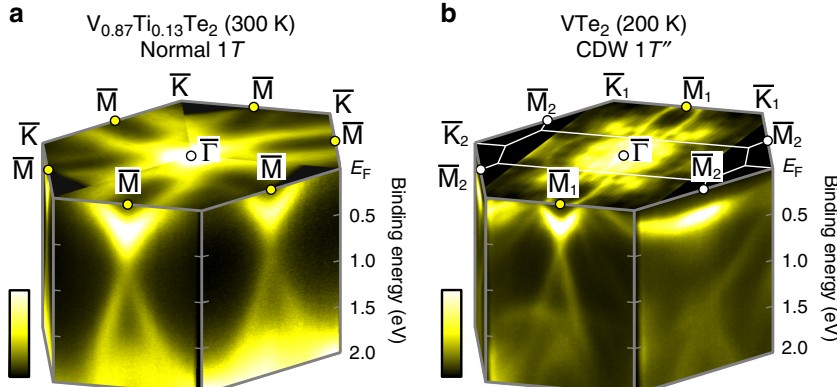

**Fig. 2 Overview of Fermi surface and band dispersions for normal 1$T$ and CDW 1$T''$ phases. a** Bird's eye ARPES image of 1$T$-V$_{0.87}$Ti$_{0.13}$Te$_2$ (300 K, $h\nu$ = 21.2 eV). **b** Same as **a**, but for 1$T''$-VTe$_2$ (200 K, $h\nu$ = 90 eV).

To grasp the essential electronic modification via the CDW formation, we survey the temperature- and doping-dependent ARPES results (He-discharge lamp, $h\nu$ = 21.2 eV). Note that the 1$T''$ phase inevitably contains the in-plane 120-degree CDW domains reflecting the threefold symmetry of 1$T$, and usually ARPES measurements include the signals of multiple domains (see Supplementary Note 5). Figure 3e, f respectively shows the ARPES image and corresponding EDCs of V$_{0.90}$Ti$_{0.10}$Te$_2$ in the CDW 1$T''$ phase (multi-domain, 20 K, #2). Comparing with the high-temperature 1$T$ phase (#1), we notice a very unusual flat band appearing at $E_B \sim$ 0.1–0.25 eV as denoted by the red curve in Fig. 3e, spreading over the measured momentum region. This is most well recognized in Fig. 3f as the series of EDC peaks at $E_B \sim$ 0.2 eV around the $\bar{K}_1/\bar{K}_2$ point. The blue broken curve in Fig. 3e, on the other hand, indicates the dispersive band crossing $E_F$ resembling the high-temperature normal phase (see Supplementary Note 6 for the detailed temperature-dependence). Such coexistence of flat/dispersive bands results from the mixing of CDW domains. In the pristine 1$T''$-VTe$_2$ (multi-domain, 15 K, #3) as shown in Fig. 3g, h, the similar flat band with slightly different energy and dispersion is

more clearly observed (the red broken curve), together with the 1$T$-like dispersive band (the blue broken curve). The appearance of this anomalous flat band is thus the common signature of the CDW 1$T''$ phase, which however is seemingly beyond the simple band folding and gap opening in the Fermi surface nesting scenario. The localized nature of this electronic structure will be discussed later.

**Bulk and surface band structures in 1$T$ normal phase.** Here we introduce the topological aspect that can be relevant to the 1$T$-1$T''$ CDW transition. In the normal 1$T$ phase, the band calculation suggests the band inversion involving V3$d$ and Te5$p$ orbitals at around the M and L points. The calculations of 1$T$-V$_{0.87}$Ti$_{0.13}$Te$_2$ at several $k_z$, plotted along the direction parallel to Γ-M ($k_{\Gamma M}$, see Fig. 4a) are displayed in Fig. 4b. The colors of curves show the weight of atomic orbitals depicted by a false color-scale (see Supplementary Note 3 for detailed orbital components), whereas the black broken curves are the results without SOC. Focusing on the topmost two bands at M and L, labelled as A and B, we find that their orbital characters of mainly V3$d$ (blue-like, even parity) and Te5$p_x + p_y$ (red-like, odd parity) get inverted at a finite $k_z$

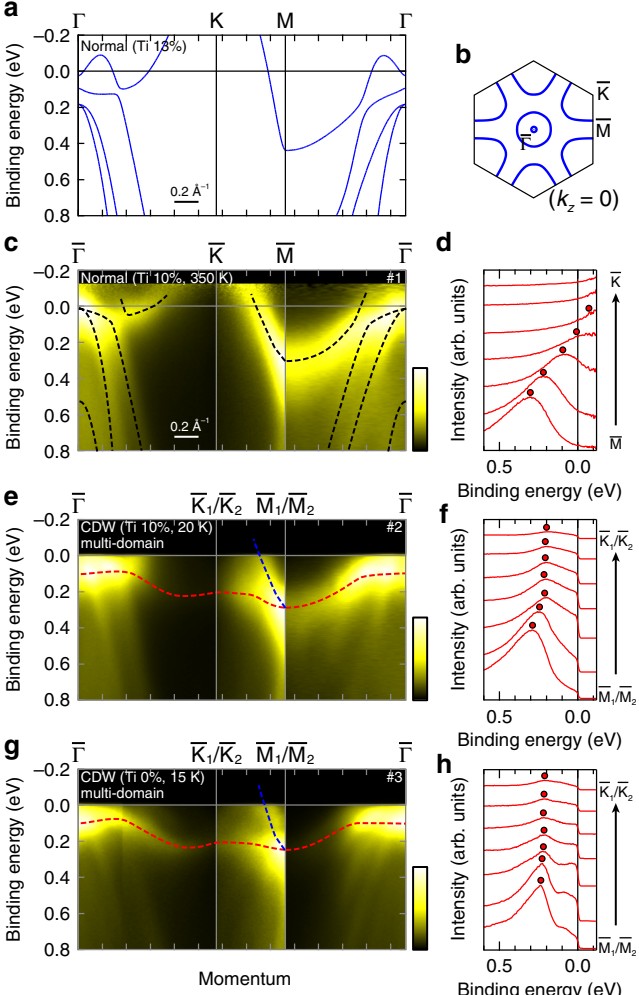

**Fig. 3 Temperature/Ti-doping dependence of electronic structures. a** The band calculation of $V_{0.87}Ti_{0.13}Te_2$ in the normal phase along the high symmetry line ($\Gamma$-K-M-$\Gamma$). **b** The corresponding Fermi surface in the $k_z = 0$ plane. **c** ARPES image of $V_{0.90}Ti_{0.10}Te_2$ in the normal phase (350 K, marked as #1 in Fig. 1f) recorded on the high symmetry lines. The data are divided by the Fermi-Dirac function convoluted with the Gaussian resolution function, to eliminate the thermal broadening of the Fermi cutoff. The broken black curves are the guide for the band dispersions, obtained by tracking the peaks of the energy and momentum distribution curves. **d** Energy distribution curves along the $\bar{M} - \bar{K}$ line for **c** (integral width: 0.05 $Å^{-1}$). **e** ARPES image of $V_{0.90}Ti_{0.10}Te_2$ in the CDW phase (20 K, #2) for the same momentum region as **c**. The red and blue curves indicate the flat bands and dispersive band near the Fermi level, obtained in a similar manner with **c**. **f** Same as **d**, but for **e** (#2). **g**, **h** Same as **e**, **f**, but for the pristine VTe₂ in the CDW phase (15 K, #3). All the ARPES data in the figure were obtained using a He-discharge lamp ($h\nu = 21.2$ eV). The data in **e–h** include the signal from all the in-plane 120-degree CDW domains.

($\sim 0.76 \, \pi/c$). In the corresponding (0 0 1) slab calculation (Fig. 4c), a new band dispersing around $E - E_F = -0.6$ eV near $\bar{M}$ can be recognized, that does not exist in the bulk calculations. It roughly follows the trajectory of the virtual crossing points of bulk bands A and B for no SOC. This is a surface state topologically protected by the band inversion at $(a^*/2, 0, k_z)$ occurring due to the moderate $k_z$ dispersions of V3$d$ and Te5$p$ bands. Indeed, in the $\bar{K} - \bar{M} - \bar{K}$ direction (Fig. 4d), this surface state shows a Dirac cone-like dispersion connecting bands A and B. Figure 4e shows the ARPES

image of normal-state $V_{0.90}Ti_{0.10}Te_2$ along $\bar{K} - \bar{M} - \bar{K}$ (350 K, $h\nu = 21.2$ eV). By carefully analyzing the EDC/MDC (see Supplementary Note 7), we can indeed quantify the bottom of bulk band A ($E_B \sim 0.30$ eV), the top of bulk band B (~0.90 eV), and the crossing point of the surface Dirac cone (DP, ~0.66 eV). We note that this bulk band A forms the triangular Fermi surfaces centered at $\bar{K}$ points in the $k_z = 0$ plane, as shown in the Fermi surface image in Fig. 4f ($V_{0.87}Ti_{0.13}Te_2$, 300 K, $h\nu = 83$ eV).

We further perform the $h\nu$-dependent ARPES to confirm the two-dimensionality of the Dirac surface states and to clarify the $k_z$-dependent bulk electronic structure that is relevant to the band inversion. Figure 4h–j displays the ARPES images of 1$T$-$V_{0.90}Ti_{0.10}Te_2$ near the $\bar{M}$ point (see the red arrow in Fig. 4g), recorded at 320 K with different photon energies, $h\nu = 63, 69,$ and 78 eV, respectively corresponding to $k_z \sim 0, \pi/2c,$ and $\pi/c$. Circle markers with vertical (horizontal) error bars represent the peak positions of EDCs (MDCs) (see Supplementary Note 7). Figure 4k displays the schematic band dispersions overlaid with the experimental peak plots extracted from Fig. 4h–j. Here we find that the bulk bands A and B clearly show the finite $k_z$-dispersions (respectively larger than 0.1 and 0.4 eV at $\bar{M}$) As can be seen in Fig. 4b, this $k_z$-dispersion is essential for the band inversion at $(a^*/2, 0, k_z)$, the origin of the topological surface state appearing around the $\bar{M}$ point. On the other hand, the Dirac surface state, that is highlighted by the overlaid orange curves in Fig. 4k, is almost independent of $h\nu$, indicating the two-dimensional nature of the topological surface state.

**Bulk and surface band structures in 1$T''$ CDW phase.** To unambiguously elucidate the anisotropic electronic structures in the CDW 1$T''$ phase, here we utilize the small spot size (typically 300 μm) of the synchrotron light beam and separately measure the in-plane CDW domains of VTe₂ (see Supplementary Note 5). Figure 5a shows the Fermi surface image of the single-domain 1$T''$-VTe₂ (200 K, $h\nu = 90$ eV). We find that the two sides of the triangular Fermi surface around $\bar{K}$ observed in the normal 1$T$ phase (Fig. 4f) are completely absent in the CDW state, and the remaining one forms the quasi-one-dimensional Fermi surface marked by the red broken curves. Figure 5b, d displays the ARPES images along representative two cuts (cut #1 and #2 as denoted in Fig. 5a), nearly along $\bar{K}_1 - \bar{M}_1 - \bar{K}_1$ and $\bar{K}_1 - \bar{M}_2 - \bar{K}_2$, respectively. Though they are originally the equivalent momentum cuts in the normal 1$T$ phase, distinctive features are clearly observed. Along $\bar{K}_1 - \bar{M}_1 - \bar{K}_1$, there are several bands crossing $E_F$ including the 1$T$-like dispersion (the black broken curve), together with the Dirac-cone-like band (the white broken curve) as clearly seen in the MDCs (Fig. 5c). On the other hand, along $\bar{K}_1 - \bar{M}_2 - \bar{K}_2$, the bands crossing $E_F$ as well as the Dirac band completely disappear (see the MDCs in Fig. 5e), and the peculiar flat band (the black broken curve in Fig. 5d) appears around $E_B \sim 0.2$–0.3 eV. These indicate that the CDW induces the drastic directional modification of electronic structure.

Here, to confirm the topological nature of the Dirac surface state in VTe₂, we performed spin-resolved ARPES on a multi-domain sample (see Supplementary Note 8 for the experimental setup). Figure 5f depicts the schematic 2D BZ with CDW multi-domains, together with the measurement region (the red arrow). Figure 5g shows the spin-integrated ARPES image with the peak plots obtained from spin-resolved spectra (15 K, $h\nu = 21$ eV, $s$ polarization). By comparing with the results from the single-domain measurement (Fig. 5b–e), these peaks can be easily assigned to either $\bar{M}_1$ or $\bar{M}_2$ side. The red (blue) and purple (cyan) triangle markers respectively represent the peak positions of spin-up (-down) spectra at $\bar{M}_1$ and $\bar{M}_2$ sides. As shown in the spin-resolved spectra in Fig. 5h, the lower branch of the Dirac cone band clearly shows the spin polarization with sign reversal at

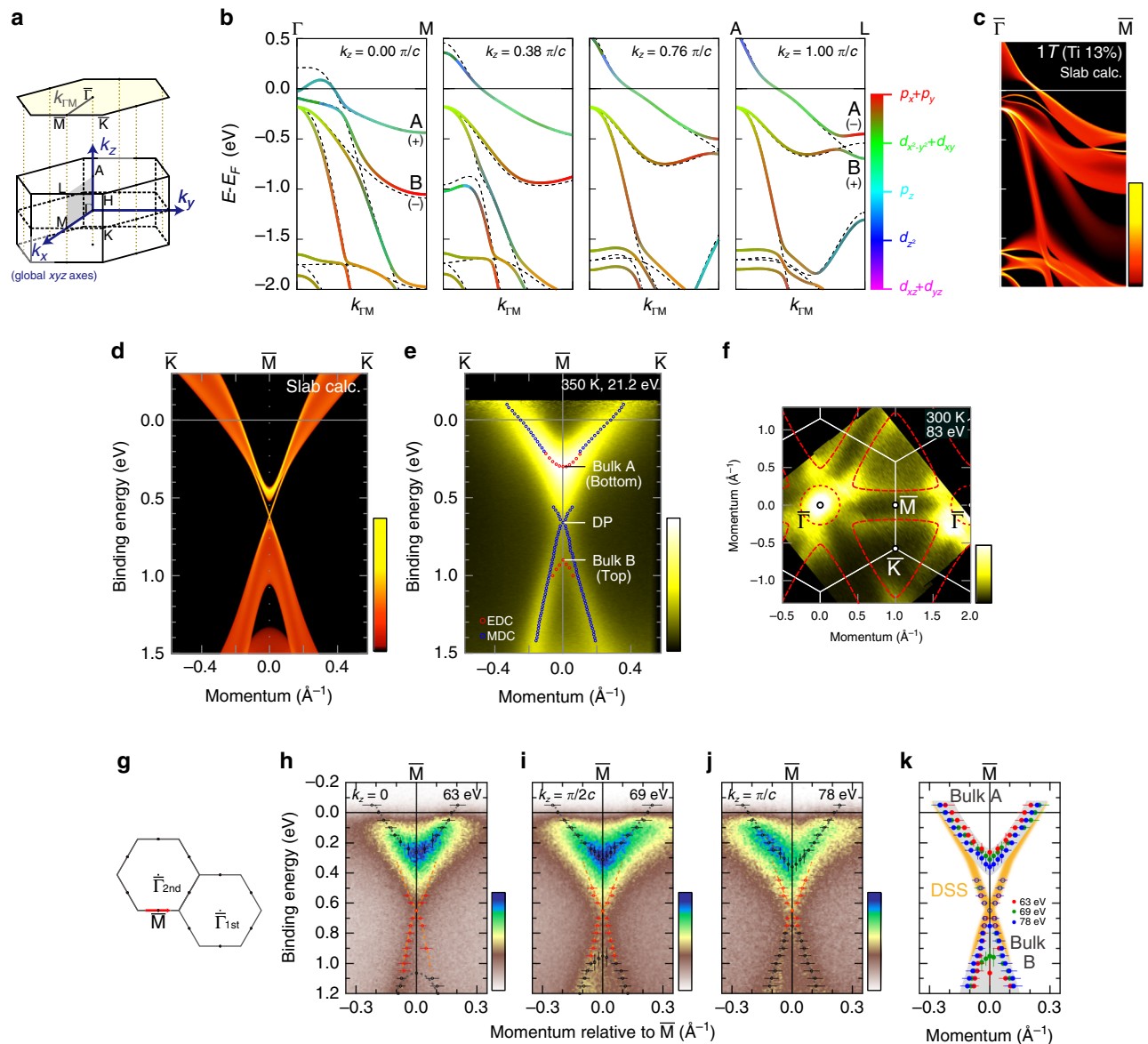

**Fig. 4 Topological character of band structures in the normal 1T phase for Ti-doped VTe₂. a** The Brillouin zone of 1T phase. $k_x$, $k_y$, and $k_z$ indicate the axes in the reciprocal space corresponding with the global $xyz$ axes (see Fig. 1a) used for the calculation in **b**. **b** The bulk band calculations of 1T-$V_{0.87}Ti_{0.13}Te_2$ along $k_{\Gamma M}$ direction at several $k_z$ values (0, 0.38, 0.76, 1.00 $\pi/c$). The color of the curves indicates the weight of orbital characters, as shown by the color-scale. The "+" and "−" mark the even and odd parity, respectively. The broken curves represent the results without spin-orbit coupling. **c**, **d** The corresponding (0 0 1) slab calculations along the $\bar{\Gamma} - \bar{M}$ (**c**) and the $\bar{K} - \bar{M} - \bar{K}$ (**d**) lines, respectively. **e** ARPES image of 1T-$V_{0.90}Ti_{0.10}Te_2$ along the $\bar{K} - \bar{M} - \bar{K}$ direction recorded using a He-discharge lamp (21.2 eV, 350 K). The red (blue) marks represent the peak positions of energy (momentum) distribution curves. **f** Fermi surface image of 1T-$V_{0.87}Ti_{0.13}Te_2$ in the $k_z = 0$ plane (circular polarized 83 eV photons, 300 K), with an energy window of ±5 meV. The red curves display the Fermi surface schematically. **g** Schematic 2D BZ. The red arrow indicates the measurement region of $h\nu$-dependent ARPES. **h–j** $h\nu$-dependent ARPES spectra of 1T-$V_{0.90}Ti_{0.10}Te_2$ with 63 eV (**h**), 69 eV (**i**), and 78 eV (**j**) photons (circular polarization, 320 K), which respectively detects at $k_z = 0$, $\pi/2c$, and $\pi/c$ plane. The circles with vertical (horizontal) error bars represent the EDC's (MDC's) peak positions. **k** Schematic band dispersion along $\bar{K} - \bar{M} - \bar{K}$ with experimentally obtained peak plots. The red, green, and blue markers are the peak plots for $h\nu = 63$, 69, and 78 eV, respectively from **h–j**. The orange curve represents the schematic of the Dirac surface state (DSS), whereas the blurred gray curves are the bulk bands (Bulk A and B).

$\bar{M}_1$ point (i.e., Dirac crossing point), similarly to the topological surface state in the topological insulators. On the other hand, the bulk flat bands around $E_B \sim 0.25$ eV at $\bar{M}_2$ side indeed show the spin degenerate character. Figure 5i is an expanded viewgraph of spin-resolved spectra near the Fermi level at emission angle $\theta = \pm 4°$. There are slight spin-up/down intensity contrasts just below $E_F$ ($E_B \sim 0.05$ eV), which should be corresponding to the upper branch of the surface Dirac cone band at $\bar{M}_1$ side.

Similarly to the normal phase, $h\nu$-dependent ARPES measurement is performed on the CDW state VTe₂ (15 K, multi-domain) to clarify the $k_z$-dependence of electronic structures. Figure 5j–l displays the ARPES image near the $\bar{M}_1/\bar{M}_2$ points (see the red arrow in Fig. 5f), recorded with $h\nu = 54$, 61.5, and 69 eV, respectively. Again by comparing with the single-domain measurement, we can assign the peaks to either $\bar{M}_1$ or $\bar{M}_2$ side (see Supplementary Note 9). Figure 5m, n respectively shows

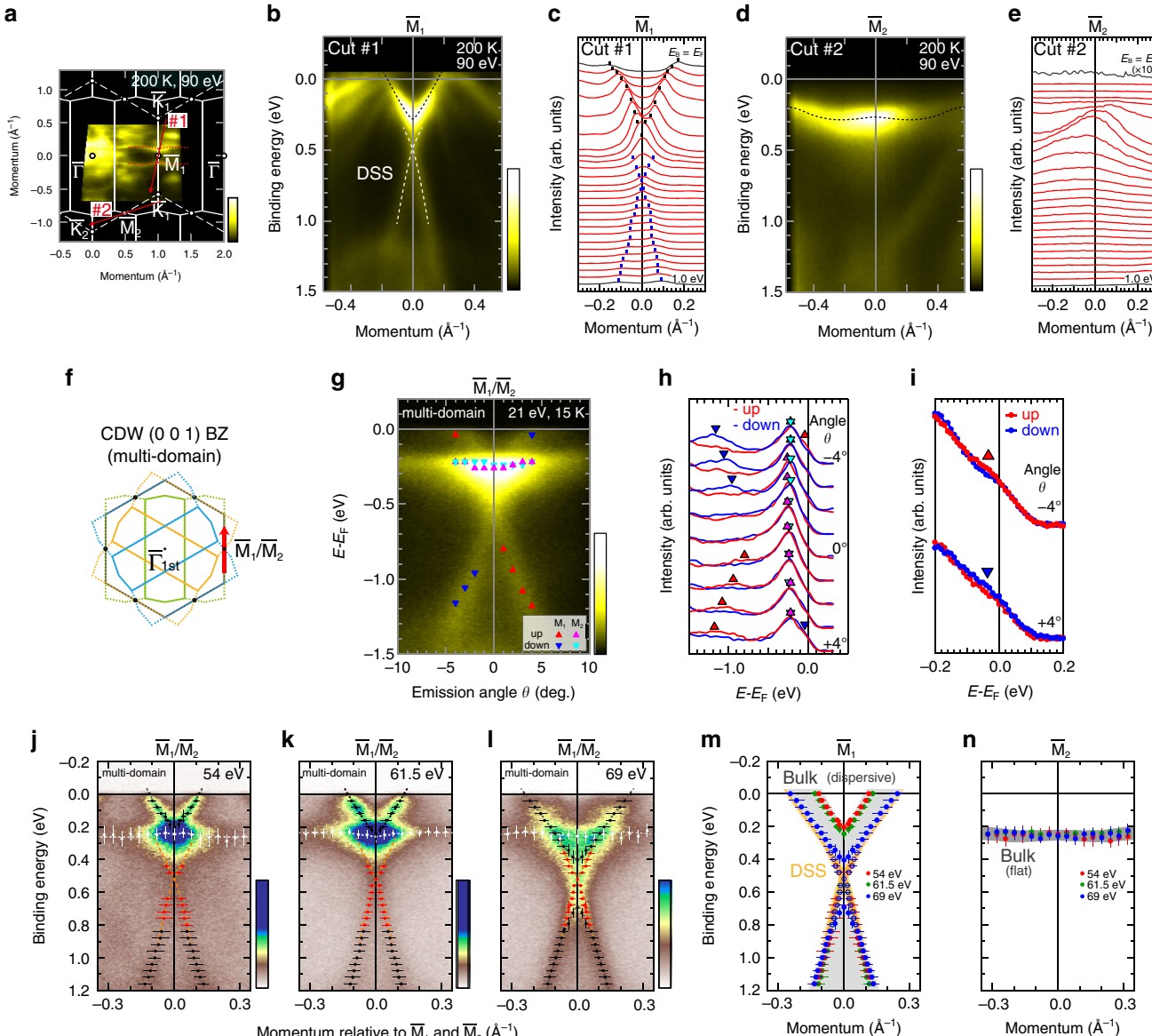

**Fig. 5 Anisotropic electronic structures in the CDW 1T″ phase for VTe₂. a** Fermi surface image of single-domain VTe$_2$ (circular polarized 90 eV photons, 200 K). White broken lines and full lines indicate the 2D BZ in the 1T and 1T″ phase, respectively. The red dotted curves denote the quasi-one-dimensional Fermi surface peculiar to the 1T″ phase. **b** ARPES image along cut #1 (as depicted in **a**), nearly along $\bar{K}_1 - \bar{M}_1 - \bar{K}_1$ direction. The black and white dotted curves respectively trace the dispersive bulk band and the Dirac surface state similar to 1T. **c** MDCs for **b**, ranging from $E_B = 0$ to 1.0 eV (integral width: 0.02 eV). The black (blue) makers represent the MDC peak positions for the dispersive bulk band (Dirac surface state). **d**, ARPES image along cut #2, nearly along $\bar{K}_1 - \bar{M}_2 - \bar{K}_2$. The black dotted curve traces the flat band peculiar to 1T″. **e** MDCs for **d**. The MDC at $E_F$ is multiplied by 10. **f** Schematic 2D BZ with mixed in-plane CDW domains. The red arrow qualitatively follows the measurement direction of spin-resolved and $h\nu$-dependent ARPES. **g** ARPES image with the peak plots of spin-resolved spectra (s polarized 21 eV photons, 15 K). The domains are not separated. Red (blue) triangle markers indicate the peak positions of spin-up (-down) spectra of $\bar{M}_1$ side bands, whereas purple (cyan) for $\bar{M}_2$ side. **h** Spin-resolved spectra around $\bar{M}_1/\bar{M}_2$ points. **i** Spin-resolved spectra near $E_F$ at $\theta = \pm4°$. **j–l** $h\nu$-dependent ARPES images of a multi-domain sample recorded with 54 eV (**j**), 61.5 eV (**k**), and 69 eV (**l**) photons (s polarization, 15 K). Circles markers with vertical (horizontal) error bars represent the peak positions of EDCs (MDCs). The black and red markers are respectively assigned to the dispersive bulk band and the Dirac surface state at $\bar{M}_1$ side, while the white markers correspond to the flat bulk state at $\bar{M}_2$ side. **m** Schematic band dispersion along $\bar{K}_1 - \bar{M}_1 - \bar{K}_1$ with experimental band plots. The orange curve represents the schematic of the Dirac surface state, whereas the blurred gray curves are those for the bulk bands. **n** Same as **m**, but along $\bar{K}_1 - \bar{M}_2 - \bar{K}_2$.

the schematic band dispersions along $\bar{K}_1 - \bar{M}_1 - \bar{K}_1$ and $\bar{K}_1 - \bar{M}_2 - \bar{K}_2$, overlaid with the experimental peak plots extracted from Fig. 5j–l. At the $\bar{M}_1$ side, the V-shaped band with high intensity clearly shows the finite $k_z$-dispersion of >0.2 eV, together with the $h\nu$-independent Dirac surface state. These are similar to the case of normal 1T phase along $\bar{K} - \bar{M} - \bar{K}$ (Fig. 4k). On the other hand, at the $\bar{M}_2$ side, the emergent flat band shows negligible $k_z$-dependence, in contrast to the V-shaped band

observed at the $\bar{M}_1$ side. With this we can conclude that the electronic state at the $\bar{M}_2$ side has unusually localized nature, which should give rise to the dissolution of the band inversion along $k_z$ direction.

**Picture of CDW state based on orbital bonding.** To grasp the essential feature of the CDW state, here we introduce the local orbital picture. For simplicity, we adopt an orthogonal octahedral

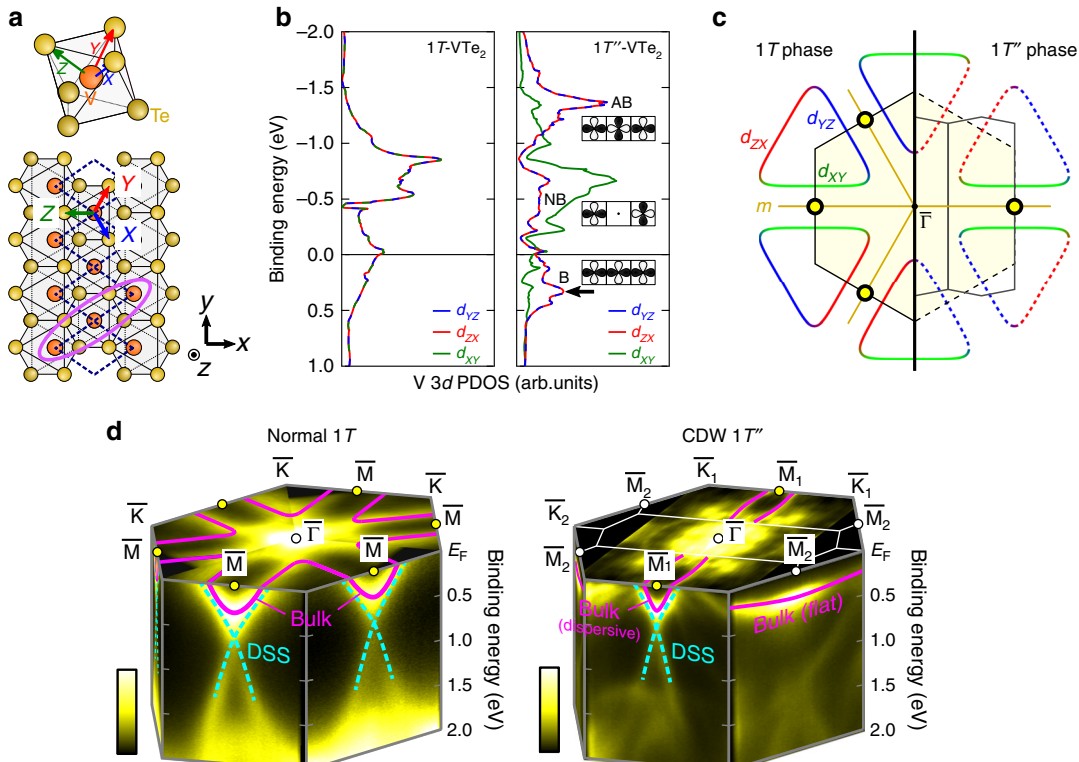

**Fig. 6 Strongly orbital-dependent electronic modification across the phase transition. a** Definition of the local orthogonal coordination ($XYZ$), set along the VTe$_6$ octahedron. Here we set $Z$ as the direction perpendicular to the vanadium's double zigzag chains in the CDW 1$T''$ phase. The pink oval shows the formation of trimer-like $d_{YZ}$ bonding. **b** Calculated partial density of states (PDOS) for V3$d$ $t_{2g}$ orbitals, $d_{XY}$, $d_{YZ}$, and $d_{ZX}$. The left (right) panel shows the result for 1$T$ (1$T''$) phases. "B", "NB", and "AB" indicate the bonding, nonbonding, and antibonding states in 1$T''$, respectively, as schematically drawn in the right panel. The black arrow represents the energy position of the flat band. **c** Schematic drawings of Fermi surfaces around the BZ boundaries, for 1$T$ (left) and 1$T''$ (right) phases. The color of the curves indicates the orbital characters. The yellow circle markers indicate the location of the Dirac crossing points where the Dirac surface states emerge. The mirror planes are depicted as "$m$". **d** Summary of band structures for 1$T$ and 1$T''$ phases made by the obtained ARPES images (1$T$-V$_{0.87}$Ti$_{0.13}$Te$_2$ (300 K, $h\nu$ = 21.2 eV) and 1$T''$-VTe$_2$ (200 K, $h\nu$ = 90 eV)). The purple and dotted cyan curves represent the bulk bands and Dirac surface states, respectively.

$XYZ$ coordination by considering the VTe$_6$ octahedron as shown in Fig. 6a, and focus on V3$d$ $t_{2g}$ ($d_{XY}$, $d_{YZ}$, $d_{ZX}$) orbitals that dominate the density of states near $E_F$ (see Supplementary Note 10). Note that this is different from the global $xyz$ setting adopted in Fig. 4b, where $z$ corresponds to the stacking direction. Here we choose $Z$ as the V-Te bond direction that is perpendicular to the vanadium's chain direction $b_m$.

Figure 6b shows the calculated partial density of states (PDOS) for vanadium $d_{XY}$, $d_{YZ}$, and $d_{ZX}$ orbitals. In 1$T$ (Fig. 6b left), they are naturally degenerate reflecting the trigonal symmetry. In 1$T''$ (Fig. 6b right), on the other hand, $d_{YZ}/d_{ZX}$ and $d_{XY}$ orbitals have strikingly different distributions. We can classify the PDOS for 1$T''$ into three major parts, lower, middle, and upper, respectively lying around $E_B \sim 0.4$, $-0.5$, and $-1.3$ eV. They are indicative of bonding (B), nonbonding (NB), and antibonding (AB) bands, arising from the trimerization-like displacements of three adjacent vanadium atoms formed by the σ-bonding of $d_{YZ}/d_{ZX}$ orbitals[23,33], as marked by the pink oval in Fig. 6a for $d_{YZ}$. Through the $d_{YZ}/d_{ZX}$ trimerization, the remaining $d_{XY}$ stays relatively intact, thus its PDOS mostly contributes to the middle band. Here, the $d_{YZ}/d_{ZX}$ bonding band makes a peak in PDOS at around $E_B \sim 0.3$ eV, corresponding to the flat bands lying around $\bar{\Gamma} - \bar{K}_2 - \bar{M}_2$ (see Supplementary Note 10 for the band dispersions). The experimentally observed flat band in Fig. 5d should be thus reflecting the localized nature of $d_{YZ}/d_{ZX}$ trimers.

Such orbital bonding picture is also useful for the intuitive understanding of Fermi surfaces at BZ boundary. Figure 6c

depicts the schematics of the Fermi surfaces with the locations of the Dirac crossing points (yellow circle makers at $\bar{M}$ points). According to the calculation in 1$T$, the three sides of the triangular Fermi surface mainly consist of $d_{YZ}$, $d_{ZX}$, and $d_{XY}$ orbitals, respectively (see Supplementary Note 10). This can be regarded as the virtual combination of one-dimensional Fermi surfaces formed by the σ-bonding of $d_{YZ}$, $d_{ZX}$ and $d_{XY}$ orbitals, raised as the basic concept of the hidden nesting scenario in CdI$_2$-type TMDCs[23] and purple bronzes $A$Mo$_6$O$_{17}$ ($A$ = Na, K)[25–27]. In the CDW 1$T''$ phase, the two sides of the Fermi surfaces composed of $d_{YZ}/d_{ZX}$ orbitals ($\bar{M}_2$ sides) turn into the localized flat bands as a consequence of the vanadium trimerization, while the one composed of $d_{XY}$ remains intact. This explains the quasi-2D to quasi-1D change of the triangular Fermi surface at BZ boundary as obtained in the present result.

## Discussion

Figure 6d summarizes the observed anisotropic reconstruction of the bulk and topological surface states occurring around the BZ boundaries by the CDW formation. As discussed, the band structures observed in the normal 1$T$ phase are basically in a good agreement with our calculations, indicating the triangular hole Fermi surfaces around the $\bar{K}$ points. At $\bar{M}$ points, the Dirac-type topological surface states exist in addition to the V-shaped bulk bands. We attribute this to the band inversion of V3$d$ and Te5$p$ orbitals occurring at ($a*/2$, $0$, $k_z$), which can be traced

experimentally (Fig. 4k). In the CDW state, on the other hand, drastic changes show up in the electronic structure. The originally triangular hole Fermi surface around the $\bar{K}$ point loses its two sides, and the remaining one forms the one-dimensional-like Fermi surface at the $\bar{M}_1$ side. We note that this corresponds to the vanadium zigzag chain direction. At the $\bar{M}_2$ side, the V-shaped bulk band transforms into the flat band at $E_B \sim 0.25$ eV, reflecting the vanadium trimerization. The original Dirac surface state survives at $\bar{M}_1$, but can be no longer seen at $\bar{M}_2$. The disappearance of the Dirac surface state can be ascribed to the absence of the band inversion at $\bar{M}_2$, where the $k_z$-dependence of the bulk state is lost due to the flat band formation (Fig. 5n), in contrast to the $\bar{M}_1$ side (Fig. 5m). Thus, in this system, the CDW accompanying the metal trimerization plays a crucial role in selectively switching off the band inversion and corresponding topological surface state.

Such a drastic modification of band inversion by CDW can be further explained based on the change in crystal symmetry. In the high-temperature 1T phase ($P\bar{3}m1$), as mentioned earlier, the Dirac cones appear at the $\bar{M}$ points, due to the band inversions involving the Te5$p$ and V3$d$ orbitals occurring along M-L lines. Here, the M-L line resides in a mirror plane, which prohibits the hybridization between the band consisting of Te5$p$ antibonding orbitals (odd with respect to the mirror plane) and that derived from the predominantly V3$d$ with finite Te5$p$ bonding orbitals (even with respect to the mirror plane), without the help of SOC. Eventually, the non-relativistic bands cross at $k_z \sim 0.76\ \pi/c$ along M-L depicted as the broken curves in Fig. 4b, thus causing the band inversion, due to the $k_z$ dependent energy eigenvalues. Only by the SOC, they get mixed and make a gap ($\sim$200 meV) around the crossing point. In the low-temperature 1T″ phase ($C2/m$), on the other hand, while the $\bar{M}_1$ point keeps the similar situation with the $\bar{M}$ point in high-temperature phase, $\bar{M}_2$ no longer resides in the mirror plane. At $\bar{M}_2$, the V3$d$ and Te5$p$ (bonding, antibonding) can now significantly mix and form the well hybridized flat band with no $k_z$ dependence, leading to the disappearance of the band inversion. This can be also confirmed by evaluating the orbital components in our bulk calculation (see Supplementary note 11). It indicates the significant $d_{YZ}/d_{ZX}$-$p_Z$ hybridization (here $X, Y, Z$ represent the octahedral setting, see Fig. 6a) forming the flat band at $E_B \sim 0.3$–$0.4$ eV. In contrast, the mirror plane remains for $\bar{\Gamma} - \bar{M}_1$, thus the band inversion and the related Dirac surface state can sustain.

Finally, we would like to mention that the topological character of materials should be determined by considering the detailed band structures for the whole system and quantifying the Berry curvature and topological invariants ($Z_2$ index, (spin) Chern number, etc.). The difficulty in the present case, however, is that the accuracy of the band calculation on 1T″ phase is still insufficient. Though the qualitative features (such as the quasi-one-dimensional Fermi surface and the flat band formation) are quite well reproduced (see Supplementary Note 12 for the band unfolding calculation), the detailed band structures including their relative energy positions are not in good enough agreement. This makes it difficult to fully discuss the topology which sharply depends on the crossings of band energy levels, particularly on its modification across the CDW transition. In this viewpoint, we would rather like to stress that our present results and microscopic analysis are offering one way to pursue the topological character of a complex material that is still difficult to predict.

In conclusion, we systematically clarified the bulk and surface electronic structures of (V,Ti)Te$_2$ by utilizing ARPES and first-principles calculations. We revealed that the strongly orbital-dependent reconstruction of the electronic structures occurs through the CDW formation, giving rise to the topological

change in particular bands. Considering that CDWs are often flexible to external stimuli[7,10–12], we can also expect to manipulate the CDW-coupled topological state. The effect of thinning the crystals down to atomic thickness is also worth investigating, which may give rise to hidden CDW states[10] and electronic phase transitions[34,35]. The combination of CDW ordering and topological aspects may lead to the new stage of manipulating the quantum materials.

## Methods

**Sample preparation.** High-quality single crystals of V$_{1−x}$Ti$_x$Te$_2$ were grown by the chemical vapor transport method with TeCl$_4$ as a transport agent. Temperatures for source and growth zones are respectively set up as 600 °C and 550 °C. The Ti concentrations ($x$) were characterized by energy dispersive x-ray spectrometry (EDX) measurements. To access both 1T and 1T″ phases, we used V$_{0.90}$Ti$_{0.10}$Te$_2$, V$_{0.87}$Ti$_{0.13}$Te$_2$, and VTe$_2$ samples with the transition temperature of ~280 K, ~250 K and ~475 K, respectively.

**ARPES measurements.** ARPES measurements were made at the Department of Applied Physics, The University of Tokyo, using a VUV5000 He-discharge lamp and a VG-Scienta R4000 electron analyzer. The photon energy and the energy resolution are set to 21.2 eV (HeIα) and 15 meV. The spot size of measurements was ~2 × 2 mm$^2$. The data in Figs. 2a, 3c–h and 4e were obtained with this condition.

Photon-energy-dependent and domain-selective ARPES measurements were conducted at BL28A in Photon Factory (KEK) by using a system equipped with a Scienta SES2002 electron analyzer. Photon-energy-dependent measurements on 1T-V$_{0.90}$Ti$_{0.10}$Te$_2$ and 1T″-VTe$_2$ were performed respectively using circular and s-polarized light ($hν = 45$–90 eV). The energy resolution was set to 30 meV. The relation between the incident light energy and the detected $k_z$ value is estimated from the experimentally obtained work function (~3.7 eV) and inner potential (~8.8 eV). The data in Figs. 4h–j and 5j–l were obtained in this condition. Domain-selective measurements on 1T″-VTe$_2$ were performed at 200 K using a 90 eV circularly polarized light with a spot size of ~100 × 300 μm$^2$. The energy resolution was set to 50 meV. The data in Fig. 5b, d were obtained from the different in-plane domains of a single sample, by translating the sample without changing its orientation. The Fermi surface images in Fig. 4f, 5a were also obtained with this instrument.

Spin-resolved ARPES measurements were performed at Efficient SPin Resolved SpectroScOpy end station attached to the APPLE-II-type variable polarization undulator beamline (BL-9B) at the Hiroshima Synchrotron Radiation Center (HSRC)[36]. The analyzer of this system consists of two sets of very-low-energy electron diffraction spin detectors, combined with a hemispherical electron analyzer (VG-Scienta R4000). Measurements of 1T″-VTe$_2$ were performed using a s-polarized 21 eV light at 15 K. In the spin-resolved measurements, we set the energy and angle resolutions to 120 meV and ±1.5 deg, respectively. We adopted the Sherman function $S_{eff} = 0.25$ for analyzing the obtained data. The data in Fig. 5g–i were obtained with this condition.

In all the ARPES experiments, samples were cleaved at room temperature in-situ, and the vacuum level was better than $5 \times 10^{−10}$ Torr through the measurements. The Fermi level of the samples were referenced to that of polycrystalline golds electrically connected to the samples.

**Band calculations.** The relativistic electronic structure of 1T-V$_{0.87}$Ti$_{0.13}$Te$_2$ was calculated within the density functional theory (DFT) using the Perdew-Burke-Ernzerhof (PBE) exchange-correlation functional corrected by the semilocal Tran-Blaha-modified Becke-Johnson potential, as implemented in the WIEN2k package[37]. The effect of Ti doping was treated within the virtual crystal approximation[38]. The BZ was sampled by a 20 × 20 × 20 $k$-mesh and the muffin-tin radius $R_{MT}$ for all atoms was chosen such that its product with the maximum modulus of reciprocal vectors $K_{max}$ becomes $R_{MT}K_{max} = 7.0$. To describe the surface electronic structure, the bulk DFT calculations were downfolded using maximally localized Wannier functions[39,40] composed of V3$d$ and Te5$p$ orbitals, and the resulting 22-band tight-binding transfer integrals implemented within a 200-unit supercell. For the pristine 1T- and 1T″-VTe$_2$, the DFT electronic structure calculations were carried out by OpenMX code (http://www.openmx-square.org/), using the PBE exchange-correlation functional and a fully relativistic $j$-dependent pseudopotentials. We adopted and fixed the crystalline structures of 1T- and 1T″-VTe$_2$ reported in ref. [20,41] and sampled the corresponding BZ by a 8 × 8 × 8 $k$-mesh.

## Data availability

The datasets that support the findings of the current study are available from the corresponding author on reasonable request.

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

## Acknowledgements

The authors thank N. Katayama for fruitful discussions. We also acknowledge H. Masuda and A. H. Mayo for their assistance with EDX measurements. N.M. acknowledges the support by the Program for Leading Graduate Schools (ALPS). Y.S., M.K., and T.So acknowledge the supports by the Program for Leading Graduate Schools (MERIT). Y.S. and T.So acknowledge the supports by Japan Society for the Promotion of Science through a research fellowship for young scientists. The spin-resolved ARPES experiments ware performed under HSRC Proposals Nos. 16AG050 and 16BG040. This work was partly supported by CREST, JST (No. JP-MJCR16F1, No. JPMJCR16F2) and the JSPS KAKENHI (No. JP17H01195, No. JP19H05826).

## Author contributions

N.M., T.So, M.S., T.Sh., and K.I. carried out (S)ARPES measurements. M.K., H.T., H.S., and S.I. carried out the crystal growth and characterization. M.S.B., Y.S., and Y.M. carried out the calculations. K.H. and H.K. shared the ARPES infrastructure at Photon Factory, KEK, and assisted with measurements. K.T., K.M., and T.O. shared the SARPES infra-structure at the Hiroshima Synchrotron Radiation Center and assisted with measure-ments. N.M. and K.I. analyzed (S)ARPES data and wrote the paper with inputs from Y.S., M.S.B., S.I., and Y.M. K.I. conceived and coordinated the research.

## Competing interests

The authors declare no competing interests.
