## [Peer Review File · Nature Communications]

Reviewers' comments:

Reviewer #1 (Remarks to the Author):

This manuscript provides a comprehensive characterization (both experimental and theoretical) of the evolution in the electronic structure of the transition-metal dichalcogenide (TMD) VTe₂ through its structural/CDW transition. This compound has been known to display a peculiar structural transition from the octahedral 1T configuration to a trimerized distorted phase at around 475 K. The nature of the distorted phase is interesting, first, because it is associated with a dimensional crossover from 2D to a more 1D-like electronic behavior arising from the quasi-1D trimerized chains of the transition metal ions. Second, the origin of the instability is related to the concept of "hidden nesting" seen in other closely related TMDs, as well as in the so-called purple bronzes, but which remains largely unexplored. Finally, it is a Weyl semimetal in the 1T phase whose fate upon the structural transition is given particular scrutiny here.

The fact that the authors study different samples of the family V_{1-x}Ti_{x}Te_{2} is quite important as it allows them to acquire the ARPES spectra on either of the 1T and 1T' phases on different samples, as well as probing the two phases on the same sample. This supports the assumption that the ARPES spectrum of the pristine compound VTe₂ will essentially be the same as the one shown in Fig. 2c.

In addition to providing a very careful and comprehensive analysis of all the ARPES data as a function of doping and temperature, the manuscript provides also a careful analysis on the underlying microscopic changes taking place through the transition. This analysis is anchored on equally detailed DFT calculations performed for both 1T and 1T' phases, as well as for doped and undoped samples. I find the various aspects of this discussion and interpretation very compelling and clear.

In my view, these are experimental and theoretical results of very good quality and the manuscript conveys a very consistent picture of the underlying physics. The possibility of manipulating topologically non-trivial electronic states on-demand is an exciting prospect, which this manuscript initiates by demonstrating one explicit example where that occurs due to a structural transition that can be induced with external perturbations. It therefore shows that the goal of achieving tunable topological states (at least as far as suppressing them) via tunable structural transitions is quite realistic. I therefore think that the manuscript meets in principle the criteria for publication in Nature Communications.

Notwithstanding, the authors should clarify/correct the following points in a revised version.

1) Page 5, line 6 -- "It is characterized by the circular and triangular hole Fermi surfaces around Γ and K, respectively". The topmost band in Fig. 2a intersects E_f 3 times along ΓK (or ΓM). This means that the statement that there is a hole pocket centered at Γ is, strictly speaking, not accurate (in fact, from Fig. 2a one could say there is a tiny electron pocket at Γ). Likewise, the energy contour at $k_z=0$ in Fig. 2b should show an additional tiny circle centered at Γ . Why is it not there and, instead of 3, we have only 2 contours in Fig. 2b.

Is it a matter of resolution? I understand this is a somewhat minor aspect that has no impact on the conclusions and interpretation of the underlying picture, but it would be good to clarify what currently looks like a small inconsistency.

2) Page 9, line 18/19 -- When discussing the hidden nesting scenario, I suggest that the authors refer

not only to TMDs but also to the case of purple bronzes, where that hidden nesting is crucial to describe the electronic structure and ARPES measurements. In this regard, the authors could cite, in addition to reference 23, the following additional references:

- M. H. Whangbo et al., Hidden fermi surface nesting and charge density wave instability in low-dimensional metals. *Science* 252, 96 (1991).

- D. Mou et al., Discovery of an Unconventional Charge Density Wave at the Surface of $KO_9Mo_6O_{17}$. *Phys. Rev. Lett.* 116, 196401 (2016).

- L. Su, C.-H. Hsu, H. Lin, V. M. Pereira, Charge Density Waves and the Hidden Nesting of Purple Bronze $KO_9Mo_6O_{17}$. *Phys. Rev. Lett.* 118, 257601 (2017).

3) Page 12, line 10 -- Please add a citation to the relevant references describing the "virtual crystal approximation".

4) Neither the main text nor supplementary sections S10, S11 explain how were the DFT calculations performed for the $1T'$ distorted phase. Were the distorted positions fixed "by hand" or allowed to relax in a supercell? If they were fixed, were experimental values used for that or, if not, how were the distorted equilibrium positions obtained? Please include a brief statement describing that procedure in the "Methods - Band Calculations" section, and provide additional details in Supplementary section S10.

5) Supplementary section S11, line 5 -- Replace "reproduce the ARPES results" by "reproduce the ARPES results for a single domain region" to emphasize that the comparison is meaningful only in the case of a single-domain region.

6) Supplementary section S11, line 6 -- "the surface state observed in ARPES measurement (Fig. 4b) does not appear in this bulk calculation". I understand that the theoretical results shown in Fig. S11 represent the unfolded spectral function of the bulk system and, consequently, surface states cannot appear there. However, as the authors do not show nor mention it anywhere in the main or supplementary texts, it is still unclear whether the calculations for the pristine $1T'$ case reveal (or not) a band inversion and the corresponding surface states, if one calculates a *slab* projection of the spectral function analogously to what is done in Fig. 3c (main text). In other words, Figs. 3a-c show both the band inversion and surface state in the calculation done for the 13% doped case. The question is what is the result of the same calculation for the undoped $1T'$ case? Is there a surface state and Weyl point near M_1 , or do all 3 equivalent Weyl points disappear? Does the result depend on the doping level used in the calculations? The authors should clarify this point because, if all the Weyl points disappear in the calculation of pristine $1T'$, that would be in contrast with the experimental result (where the Weyl point at M_1 remains), and must be carefully justified.

Reviewer #2 (Remarks to the Author):

The manuscript by Mitsuishi et al. reports an extensive ARPES investigation of the Charge Density Wave (CDW) transition in the $V_xTi_{1-x}Te_2$ ($x=0; 0.1$) transition metal dichalcogenide. Such material

hosts topological surface states (TSS) and the authors investigate the interplay between the CDW transition and the topology of the material, with a particular focus on its band inversion. This is a very interesting point because it would open the possibility to manipulate the topological phase of materials by means of CDW. The presented results are interesting, the quality of the performed research is of high level, the presentation is clear and the paper very well written. However, the presented interpretation is not conclusive in that sense. The authors states that some of the TSS are destroyed by the CDW order as it is shown by the ARPES investigation (fig. 4.b-e and fig 5.d). Moreover, the Spin-ARPES in the supplementary (section S8) shows that the remaining Dirac cone (across the M1 high symmetry point) preserves its typical spin texture. As the topology of a material is a property of the system as a whole, the data seems to demonstrate that the topology is conserved across the CDW transition and that the reorganisation of the Dirac cones occurs to follow the changed structural symmetry (C3 symmetry is lifted in the monoclinic 1T" phase). This is 'the' focal point/question of the paper and needs to be addressed more convincingly, or it should be much more elaborated.

Moreover, some further points need clarification, namely:

1. why the t_{2g} orbitals (d_{xy} d_{yz} d_{xz}) are discussed mostly to explain the removal of band inversion in the CDW phase. Looking at fig.3.a it seems that the band inversion mainly involves the p_x-p_y Te orbitals and the d_{x²-y²}-d_{xy} orbitals of V (or I'm confused by the similar colour code red-pink?). According to the authors the d_{xy} orbital is unaffected by the CDW transition, I therefore think that a discussion of the effects of CDW on the d_{x²-y²} V and p_x-p_y orbitals should be included.

2. In the text and figures three coordinate systems are chosen: crystal directions, the global xyz and XYZ centred on the V atom of the VTe₆ octahedron. I found confusing for the reader the presence of both xyz and XYZ, tough. It would be possible to adopt only one coordinate system? Otherwise, I would suggest to indicate both the systems e.g. in fig.5.a.

3. Transport data fig. S2.b Are these data necessary? I found them a bit confusing when compared with the rich ARPES dataset. For example, looking at x=0.1 transport curve, the anomaly occurs at ~400K while from the ARPES data (fig. 2, fig. 3 and supplementary section S5) it is stated that the CDW transition starts at ~250K as stated in the main text. Can the authors comment a bit more on this? Another point not completely clear to me is the apparent discrepancy between transport and ARPES data in the CDW phase: ARPES spectra show strong localisation of the electronic bands (across the M2 high symmetry point) and the arise of almost 1D electronic state, while the transport shows metallic behaviour with localisation only for very low temperature (<20K). Can the authors clarify this point?

4. Spin-ARPES data fig. S8.d As the signal is faint, I would suggest to avoid the arb. units and to indicate the counts of the I_y⁺,⁻ spectra.

A curiosity on the transport data: I am wondering why the anomaly is fainter for higher x values. Is it due to disorder introduced by the doping or it depends on the hole concentration introduced by the Ti-doping?

Reviewer #3 (Remarks to the Author):

The manuscript by Mitsuishi et al. reports changes in topological surface states in doped VTe₂, due to large changes in the crystal structure due to the formation of a 3x1 CDW. In principle, this is quite an interesting finding, although the way the paper is initially presented, it seems to imply (incorrectly)

that the CDW transitions completely changes the topological character of the material which is not the case. Instead, some of the DSSes are removed due to CDW formation, while other DSSes remain present (in the ungapped parts of the Fermi surface). This is still an interesting finding, although not as remarkable as if the entire topological character of the material changed.

The presentation of the data showing the DSSes themselves is not particularly clear, particularly in Figures 4g-h, and Figures 3g-h. I actually found that some of the data in the SI (Figs S5 and S7) seem to show the DSSes much more clearly. I do believe that the authors' main identification of the DSSes in both the 1T and 1T' phases. In addition, there is some discussion of the spin-resolved results even in the abstract, yet none of the spin-resolved measurements actually show up in the main manuscript, only the supplemental Figure S8; authors should consider moving this to the main text. Granted, the spin-resolved data is not of the best quality, although the results from the lower branch of the DSS seem relatively clear.

Furthermore, the DFT calculations in Figure 3a-c in the normal state structure appear to indicate band inversion, supported by the appearance of a DSS in the slab calculation in 3c. This seems to be sensible and consistent with the ARPES data in the normal state. What about the DFT calculations for a 3x1 supercell in the 1T' phase? Do they likewise support the author's observations of a DSS at certain k-points and absence on the gapped parts of the FS? I see in the SI that they have produced DFT calculations of the 1T' phase but am surprised that they have not produced a similar analysis. Also (a more minor question), why is a Ti doping of 13% selected (as it does not appear to correspond to any of the doping used), nor is a supercell used in the calculation?

The organization and presentation of the paper is jumbled, disorganized, and confusing. There is a lot of switching back and forth between different data sets (single domain, domain-averaged, different photon energies, etc...). It would be much more clear-cut to just have all the data taken for single domain, since the domains appear to be rather large (many hundreds of microns - this is not a technically challenging experiment). As mentioned earlier, it is not evident until quite late in the manuscript (end of pg 7, beginning of pg 8) that only some DSSes disappear and others remain unchanged - this should be made clear at the outset.

In summary, I think that the report of changes in the topological DSSes due to CDW transition in doped VTe₂ is certainly an interesting one, potentially worthy of publication in Nature Communications, and I do believe the authors' identification of this phenomenon. In its current state, however, I do not think the manuscript is ready for publication unless the authors make significant changes and improvements.

Reviewer #1 (Remarks to the Author):

This manuscript provides a comprehensive characterization (both experimental and theoretical) of the evolution in the electronic structure of the transition-metal dichalcogenide (TMD) VTe₂ through its structural/CDW transition. This compound has been known to display a peculiar structural transition from the octahedral 1T configuration to a trimerized distorted phase at around 475 K. The nature of the distorted phase is interesting, first, because it is associated with a dimensional crossover from 2D to a more 1D-like electronic behavior arising from the quasi-1D trimerized chains of the transition metal ions. Second, the origin of the instability is related to the concept of "hidden nesting" seen in other closely related TMDs, as well as in the so-called purple bronzes, but which remains largely unexplored. Finally, it is a Weyl semimetal in the 1T phase whose fate upon the structural transition is given particular scrutiny here.

The fact that the authors study different samples of the family V_{1-x}Ti_{x}Te_{2} is quite important as it allows them to acquire the ARPES spectra on either of the 1T and 1T' phases on different samples, as well as probing the two phases on the same sample. This supports the assumption that the ARPES spectrum of the pristine compound VTe₂ will essentially be the same as the one shown in Fig. 2c.

In addition to providing a very careful and comprehensive analysis of all the ARPES data as a function of doping and temperature, the manuscript provides also a careful analysis on the underlying microscopic changes taking place through the transition. This analysis is anchored on equally detailed DFT calculations performed for both 1T and 1T' phases, as well as for doped and undoped samples. I find the various aspects of this discussion and interpretation very compelling and clear.

In my view, these are experimental and theoretical results of very good quality and the manuscript conveys a very consistent picture of the underlying physics. The possibility of manipulating topologically non-trivial electronic states on-demand is an exciting prospect, which this manuscript initiates by demonstrating one explicit example where that occurs due to a structural transition that can be induced with external perturbations. It therefore shows that the goal of achieving tunable topological states (at least as far as suppressing them) via tunable structural transitions is quite realistic. I therefore think that the manuscript meets in principle the criteria for publication in Nature Communications.

Notwithstanding, the authors should clarify/correct the following points in a revised version.

We thank Reviewer #1 for his/her invaluable time and effort for reviewing our manuscript. We are also happy to see that he/she highly evaluates our work and recommends the publication in Nature Communications. The reviewer also pointed out several concerns and advices, that greatly help us to improve our manuscript. Below we answer all the questions/suggestions point-by-point.

1) Page 5, line 6 -- "It is characterized by the circular and triangular hole Fermi surfaces around Γ and K, respectively". The topmost band in Fig. 2a intersects E_F 3 times along ΓK (or ΓM). This means that the statement that there is a hole pocket centered at Γ is, strictly speaking, not accurate (in fact, from Fig. 2a one could say there is a tiny electron pocket at Γ). Likewise, the energy contour at $k_z=0$ in Fig. 2b should show an additional tiny circle centered at Γ . Why is it not there and, instead of 3, we have only 2 contours in Fig. 2b.

Is it a matter of resolution? I understand this is a somewhat minor aspect that has no impact on the conclusions and interpretation of the underlying picture, but it would be good to clarify what currently looks like a small inconsistency.

We apologize for the confusion. The reviewer is correct that the band calculation of $1T-V_{0.87}Ti_{0.13}Te_2$ indicates a tiny electron Fermi surface (FS) around Γ , as shown in Supplementary Information (Fig. S3b of the previous manuscript). It was omitted in previous Fig. 2b because we made the "schematic" FS based on the ARPES results, where the tiny electron FS pocket was not clearly detected. According to the calculation, the bottom energy of the electron band forming the tiny FS is ~ 20 meV below E_F . Considering that the calculation itself contains some energy error (~ 10 meV) and that the energy position of the bands can also severely depend on the slight change of atomic positions, we speculate that the corresponding band in the real $V_{0.87}Ti_{0.13}Te_2$ material is located above E_F . As he/she also mentioned, this small inconsistency does not affect our interpretation and conclusions at all. Nevertheless, simply to avoid readers' confusion, we have replaced the schematic FS (previous Fig. 2b) with the calculated FS at $k_z = 0$ as shown in Fig. R1 and modified the statement correspondingly.

(On page 5, line 25)

(We note that the tiny electron pocket at Γ is not clearly detected, probably reflecting the finite energy mismatch of the bands compared to the calculation.)

Fig. R1: The calculated Fermi surface of $1T-V_{0.87}Ti_{0.13}Te_2$ at $k_z = 0$.

2) Page 9, line 18/19 -- When discussing the hidden nesting scenario, I suggest that the authors refer not only to TMDs but also to the case of purple bronzes, where that hidden nesting is crucial to describe the electronic structure and ARPES measurements. In this regard, the authors could cite, in addition to reference 23, the following additional references:

- M. H. Whangbo et al., Hidden fermi surface nesting and charge density wave instability in low-dimensional

metals. *Science* 252, 96 (1991).

- D. Mou et al., Discovery of an Unconventional Charge Density Wave at the Surface of $K0.9Mo6O17$. *Phys. Rev. Lett.* 116, 196401 (2016).

- L. Su, C.-H. Hsu, H. Lin, V. M. Pereira, Charge Density Waves and the Hidden Nesting of Purple Bronze $K0.9Mo6O17$. *Phys. Rev. Lett.* 118, 257601 (2017).

We thank for his/her valuable suggestion. We fully agree with the reviewer and added some statement on purple bronzes together with the suggested references.

3) Page 12, line 10 -- Please add a citation to the relevant references describing the "virtual crystal approximation".

We thank the reviewer for his/her suggestion. We have added the relevant reference for the virtual crystal approximation as ref. [38] in Methods.

[38] L. Ballaiche and D. Vanderbilt, *Phys. Rev. B* **61**, 7877 (2000).

4) Neither the main text nor supplementary sections S10, S11 explain how were the DFT calculations performed for the $1T'$ distorted phase. Were the distorted positions fixed "by hand" or allowed to relax in a supercell? If they were fixed, were experimental values used for that or, if not, how were the distorted equilibrium positions obtained? Please include a brief statement describing that procedure in the "Methods - Band Calculations" section, and provide additional details in Supplementary section S10.

We thank him/her for the important comment, we missed the detailed information for the calculation. The DFT calculations for the distort $1T'$ phase were performed by using the experimental atomic positions reported in ref. [20] (K. D. Bronsema *et al.*, *J. Solid State Chem.*, 1984) without structural optimization. We have modified "Methods - Band Calculations" to explicitly provide the information.

5) Supplementary section S11, line 5 -- Replace "reproduce the ARPES results" by "reproduce the ARPES results for a single domain region" to emphasize that the comparison is meaningful only in the case of a single-domain region.

We thank the reviewer for his/her suggestion. We have replaced the relevant sentence in the revised manuscript as the reviewer suggested.

6) Supplementary section S11, line 6 -- "the surface state observed in ARPES measurement (Fig. 4b) does not appear in this bulk calculation". I understand that the theoretical results shown in Fig. S11 represent the unfolded spectral function of the bulk system and, consequently, surface states cannot appear there. However, as the authors do not show nor mention it anywhere in the main or supplementary texts, it is still unclear whether the calculations for the pristine $1T'$ case reveal (or not) a band inversion and the corresponding surface states, if one calculates a *slab*

projection of the spectral function analogously to what is done in Fig. 3c (main text). In other words, Figs. 3a-c show both the band inversion and surface state in the calculation done for the 13% doped case. The question is what is the result of the same calculation for the undoped 1T' case? Is there a surface state and Weyl point near M1, or do all 3 equivalent Weyl points disappear? Does the result depend on the doping level used in the calculations? The authors should clarify this point because, if all the Weyl points disappear in the calculation of pristine 1T', that would be in contrast with the experimental result (where the Weyl point at M1 remains), and must be carefully justified.

We thank the reviewer for his/her valuable comment. We fully agree that the slab calculation on 1T'' phase should be very helpful for the present discussion on topological properties. Nevertheless, since the 1T'' phase has a complicated crystal structure with many atoms in the whole unit cell (the number of atoms in 1T'' is 9 times of 1T), its slab calculation is technically demanding. Actually, after receiving this comment, we re-noticed its importance and had been trying the slab calculation in the similar manner with the normal 1T case until recently. Regrettably, however, it ended up with an unsuccessful attempt (did not converge). One problem is the calculation cost itself, but even if it is overcome, we still have certain difficulty in unfolding the obtained slab bands into the 1T BZ, to compare with ARPES. We have to wait for the future progress in the calculation techniques and/or power.

Besides, we also note that the bulk calculation for 1T''-VTe₂ itself is still lacking the quantitative accuracy for discussing the entire topological character. Though the qualitative parts of the ARPES results (e.g. the drastic change of Fermi surface and the flat band formation) are quite well reproduced, the energies of the flat bands for example are still in disagreement: They are located at binding energies of ~ 0.25 eV (ARPES) and ~ 0.4 eV (DFT). Since the topological character sharply depends on the relative energies (crossings and anti-crossings) of the multiple bands, they must be precisely reproduced in the calculation before discussing the realistic topological classification. Here we note that VTe₂ does not appear in the recently developed topological materials databases (<https://www.topologicalquantumchemistry.com/>, <http://materiae.iphy.ac.cn/>, started to operate from early 2019) as well, possibly due to the complex structure as mentioned above. In this viewpoint, we would rather like to stress that our present experimental results and the microscopic discussion offer an important key for exploring the topological matters which are still too complex to predict in the present stage.

To clearly state this situation, we added some description as the following.

(On page 12, line 13)

Finally, we would like to mention that the topological character of materials should be determined by considering the detailed band structures for the whole system and quantifying the Berry curvature and topological invariants (Z_2 index, (spin) Chern number, etc). The difficulty in the present case, however, is that the accuracy of the band calculation on 1T'' phase is still insufficient. Though the qualitative features (such as the quasi-one-dimensional Fermi surface and the flat band formation) are quite well reproduced (see Supplementary Note 12 for the band unfolding calculation), the detailed band structures including their relative energy positions are not in good enough agreement. This makes it difficult to fully discuss the topology which sharply depends on the crossings of band energy levels, particularly on its modification across the CDW transition. In this viewpoint, we

would rather like to stress that our present results and microscopic analysis are offering one way to pursue the topological character of a complex material that is still difficult to predict.

Reviewer #2 (Remarks to the Author):

The manuscript by Mitsubishi et al. reports an extensive ARPES investigation of the Charge Density Wave (CDW) transition in the $V_xTi_{1-x}Te_2$ ($x=0; 0.1$) transition metal dichalcogenide. Such material hosts topological surface states (TSS) and the authors investigate the interplay between the CDW transition and the topology of the material, with a particular focus on its band inversion. This is a very interesting point because it would open the possibility to manipulate the topological phase of materials by means of CDW. The presented results are interesting, the quality of the performed research is of high level, the presentation is clear and the paper very well written.

We thank the reviewer for his/her invaluable time and efforts on reviewing our manuscript. We are also happy to see the positive comments. The reviewer raised several suggestions/concerns that are helpful for improving our manuscript. In the following, we answer all the comments point-by-point.

However, the presented interpretation is not conclusive in that sense. The authors states that some of the TSS are destroyed by the CDW order as it is shown by the ARPES investigation (fig. 4.b-e and fig 5.d). Moreover, the Spin-ARPES in the supplementary (section S8) shows that the remaining Dirac cone (across the M1 high symmetry point) preserves its typical spin texture. As the topology of a material is a property of the system as a whole, the data seems to demonstrate that the topology is conserved across the CDW transition and that the reorganisation of the Dirac cones occurs to follow the changed structural symmetry (C_3 symmetry is lifted in the monoclinic $1T'$ phase). This is 'the' focal point/question of the paper and needs to be addressed more convincingly, or it should be much more elaborated.

We thank the reviewer for his/her important comment. We fully agree that the topological character should be discussed by considering the electronic structure of the whole system, not only the local change. Namely, the Berry curvature and topological invariants (Z_2 index, spin Chern number, etc) derived from the whole bands are the keys to classify the topological phases. For this, the DFT calculation is very effective for many weakly-correlated materials, as systematically done in recently developed topological materials databases (<https://www.topologicalquantumchemistry.com/>, <http://materiae.iphy.ac.cn/>, started to operate from early 2019). The difficulty in the present case, however, is that the accuracy of the band calculation on $1T'$ phase is still insufficient. Though the qualitative parts of the ARPES results (e.g. the drastic change of Fermi surface and the flat band formation) are quite well reproduced, the energies of the flat bands for example are still in disagreement: They are located at binding energies of ~ 0.25 eV (ARPES) and ~ 0.4 eV (DFT). Since the topological character sharply depends on the relative energies (crossings and anti-crossings) of the multiple bands, they must be precisely reproduced in the calculation before discussing the realistic topological classification. Partly related with this, the slab calculation on $1T'$ phase also ended up unsuccessful, as we replied to the comment by Referee #1. These prevent us from thoroughly discussing the topological character and its modification across the CDW transition. Here we note that VTe_2 does not appear in the abovementioned topological materials databases as well, possibly due to the complex structure. In this viewpoint, we would rather like to stress that our present experimental results and the

microscopic discussion show one way for exploring the topological matters which are still too complex to predict in the present stage.

Regarding the crystal symmetry, we can make the following arguments. In the high temperature $1T$ phase ($P\bar{3}m1$), the Dirac cones appear at the \bar{M} points, due to the band inversions involving the $\text{Te}5p$ and $\text{V}3d$ orbitals occurring along M-L lines. Here, the M-L line resides in a mirror plane (Fig. R2 (a)), which protects the $\text{Te}5p$ anti-bonding (odd with respect to the mirror plane) and the $\text{V}3d\&\text{Te}5p$ bonding (even with respect to the mirror plane) bands from hybridizing without the help of spin-orbit interaction. Eventually the non-relativistic bands cross at $k_z \sim 0.76 \pi/c$ and cause the band inversion due to the k_z dependent energy eigenvalues, depicted as the black broken curves in Fig. 4b (new version). Only by the spin-orbit interaction, they get mixed and make a gap (~ 200 meV) around the crossing point. In the low temperature phase, on the other hand, while the \bar{M}_1 point keeps the similar situation with the \bar{M} point in high- T phase, \bar{M}_2 no longer resides in the mirror plane. At \bar{M}_2 , the $\text{V}3d$ and $\text{Te}5p$ (bonding, antibonding) can now significantly mix and form the well hybridized flat band with no k_z dependence, leading to the disappearance of the band inversion. This can be also confirmed in our bulk calculation (Fig. R2 (b)). It indicates that the flat band at the binding energy of 0.4 eV is formed through the significant hybridization of d_{yz}/d_{zx} and p_z orbitals (here X, Y, Z represents the octahedral setting, see Fig. R3 (a) in reply to the next comment). In contrast, the mirror plane remains for $\bar{\Gamma} - \bar{M}_1$, thus the band inversion and the related Dirac surface state can sustain.

Figure R2: (a) $(0\ 0\ 1)$ surface BZ with the schematic Fermi surfaces around the BZ boundaries, for $1T$ (left) and $1T''$ (right) phases. The yellow circle markers indicate the crossing points of the Dirac surface states. The mirror planes are depicted as “ m ”. (b) PDOS of $1T''$ -VTe₂ for $\text{V}3d$ (d_{xy} , d_{yz} , d_{zx}) and $\text{Te}5p$ (p_x , p_y , p_z). The arrows indicate the flat bands (binding energy: ~ 0.4 eV) composed of d_{yz}/d_{zx} and p_z .

In the previous manuscript, our focus on the electronic modification across CDW had been limited to the V bond trimerization picture. Following the Referee’s suggestion, in the present version, we added the abovementioned descriptions and arguments as follows. We also modified the abstract part to more clearly state that the CDW induces the *selective* change of Dirac bands. With these revisions, we believe that our discussion has

substantially improved.

(On page 11, line 21)

Here we attempt an argument on the band inversion in association with CDW based on the change in crystal symmetry. In the high-temperature $1T$ phase ($P\bar{3}m1$), as mentioned, the Dirac cones appear at the \bar{M} points, due to the band inversions involving the Te5p and V3d orbitals occurring along M-L lines. Here, the M-L line resides in a mirror plane, which prohibits the hybridization between the band consisting of Te5p anti-bonding orbitals (odd with respect to the mirror plane) and that derived from the predominantly V3d with finite Te5p bonding orbitals (even with respect to the mirror plane), without the help of SOC. Eventually, the non-relativistic bands cross at $k_z \sim 0.76 \pi c$ along M-L depicted as the broken curves in Fig. 4b, thus causing the band inversion, due to the k_z dependent energy eigenvalues. Only by the SOC, they get mixed and make a gap (~ 200 meV) around the crossing point. In the low-temperature $1T'$ phase ($C2/m$), on the other hand, while the \bar{M}_1 point keeps the similar situation with the \bar{M} point in high- T phase, \bar{M}_2 no longer resides in the mirror plane. At \bar{M}_2 , the V3d and Te5p (bonding, antibonding) can now significantly mix and form the well hybridized flat band with no k_z dependence, leading to the disappearance of the band inversion. This can be also confirmed by evaluating the orbital components in our bulk calculation (see Supplementary note 11). It indicates the significant d_{yz}/d_{zx} - p_z hybridization (here X, Y, Z represents the octahedral setting, see Fig. 6a) forming the flat band at the binding energy of ~ 0.4 eV. In contrast, the mirror plane remains for $\bar{\Gamma} - \bar{M}_1$, thus the band inversion and the related Dirac surface state can sustain.

Finally, we would like to mention that the topological character of materials should be determined by considering the detailed band structures for the whole system and quantifying the Berry curvature and topological invariants (Z_2 index, (spin) Chern number, etc). The difficulty in the present case, however, is that the accuracy of the band calculation on $1T'$ phase is still insufficient. Though the qualitative features (such as the quasi-one-dimensional Fermi surface and the flat band formation) are quite well reproduced (see Supplementary Note 12 for the band unfolding calculation), the detailed band structures including their relative energy positions are not in good enough agreement. This makes it difficult to fully discuss the topology which sharply depends on the crossings of band energy levels, particularly on its modification across the CDW transition. In this viewpoint, we would rather like to stress that our present results and microscopic analysis are offering one way to pursue the topological character of a complex material that is still difficult to predict.

Moreover, some further points need clarification, namely:

Before answering Comment #1, let us answer Comment #2 since they are closely related.

2. In the text and figures three coordinate systems are chosen: crystal directions, the global xyz and XYZ centred on the V atom of the VTe6 octahedron. I found confusing for the reader the presence of both xyz and XYZ, though. It would be possible to adopt only one coordinate system? Otherwise, I would suggest to indicate both the systems e.g. in fig.5.a.

We thank the reviewer for his/her important comment. We admit that xyz and XYZ settings had been presented in a confusing way. Basically, since VTe_2 has a strong layered nature with van der Waals stacking along z axis, the proper setting for presenting the global band structure is xyz axes as shown in Fig. 1a (new version). However, when we discuss the chemical bonding picture for vanadium trimerization mechanism, the octahedral XYZ setting (Fig. R3 (a)) makes it much clearer to describe the directional overlaps of respective orbitals (see Figs. R3 (a) and (b) for the correspondence between XYZ and xyz settings). So, both are necessary in this paper. To remove the confusion, we added some illustrations of global xyz coordinate in Figs. 1a, 4a and 6a (in new version), and carefully re-edited the text.

Fig. R3: Schematic d orbital basis and energy levels for (a) undistorted octahedral and (b) trigonally distorted geometry.

1. why the t_{2g} orbitals (d_{xy} d_{yz} d_{xz}) are discussed mostly to explain the removal of band inversion in the CDW phase. Looking at fig.3.a it seems that the band inversion mainly involves the p_x - p_y Te orbitals and the dx^2-y^2 - d_{xy} orbitals of V (or I'm confused by the similar colour code red-pink?). According to the authors the d_{xy} orbital is unaffected by the CDW transition, I therefore think that a discussion of the effects of CDW on the dx^2-y^2 V and p_x - p_y orbitals should be included.

We thank the Reviewer for his/her important comment. First, we apologize that the multi-colored band calculation in Fig.3a (Fig. 4b in new version) is not very suitable for resolving the orbital contributions quantitatively. Our intention of showing this viewgraph is to intuitively present the band inversion involving the $Te5p$ (red-like color, parity odd, p_x+p_y) and the predominantly $V3d$ (blue-like color, parity even, including $d_{x^2-y^2}+d_{xy}$, d_{z^2} , $d_{xz}+d_{yz}$, p_z) orbitals. Note that the global xyz setting is used here. To unambiguously show the k -dependent orbital components, we newly added the set of orbital-weighted band calculations in Supplementary Note 3 (see Fig. R4). On the other

hand, for discussing the band modification in the CDW phase, we are using the octahedral XYZ setting as shown in Fig. R3 (a). Here, the directional CDW induces the great change in d_{YZ} and d_{ZX} through trimerization, while d_{XY} is mostly unaffected. As can be confirmed in Figs. R3 (a) and (b), this change spreads into the modifications of $d_{x^2-y^2+d_{xy}}$, d_{z^2} , d_{xz+y_z} orbitals in the global setting.

Fig. R4: Orbital-weighted band dispersions of $1T\text{-V}_{0.87}\text{Ti}_{0.13}\text{Te}_2$ for (a) p_x+p_y , (b) p_z , (c) $d_{x^2+y^2+d_{xy}}$, (d) d_{z^2} , and (e) $d_{xz}+d_{yz}$.

At the same time, thanks to the Reviewer's comment, we noticed that the description on how the CDW affects the $\text{Te}5p$ orbitals had been completely missing, which should be crucial for discussing the band inversion. In the previous manuscript, we explained that the flat bands at \bar{M}_2 ($E_B \sim 0.4$ eV) are mainly derived from the vanadium trimerization due to the sigma bonding of the d_{YZ}/d_{ZX} orbitals. Here, as abovementioned in the mirror symmetry argument (Fig. R2 (b)), $\text{Te } p_z$ component also shows a similar distribution to the flat bands of d_{YZ}/d_{ZX} at 0.4 eV, indicating their significant hybridization. [Here, we note that d_{YZ} and d_{ZX} both hybridize with $\text{Te } p_z$ in the edge sharing VTe_6 octahedra network, whereas the overlap for d_{XY} and p_z is negligible (see Fig. R5 (c)).] Thus, we can conclude that the CDW affects the d_{YZ}/d_{ZX} orbitals through trimerization, together with their hybridizing counterpart $\text{Te } p_z$, which eventually leads to the disappearance of band inversion at \bar{M}_2 . On the other hand, at \bar{M}_1 , the d_{XY} and p_x, p_y bands, with their band inversion, survives through the CDW transition as also mentioned above.

To more clearly describe the connection among the CDW, p - d hybridization, and band inversion, we added the description in the main text and Supplementary Note 11 as the following.

Supplementary Note 11: CDW effects on $\text{Te}5p$ orbitals.

Figures R5 (a) and (b) show the calculated PDOS of $1T''$ -VTe₂, respectively for V3d (d_{xy} , d_{yz} , d_{zx}) and Te5p (p_x , p_y , p_z). They indicate that the flat band at the binding energy of 0.3 ~ 0.4 eV is formed by the hybridization of d_{yz}/d_{zx} and p_z orbitals. Here we note that d_{yz} and d_{zx} well hybridize with p_z in the edge-sharing VTe₆ octahedral network, whereas the overlap for d_{xy} and p_z is negligibly small (see Fig. R5 (c)).

Fig. R5: PDOS of $1T''$ -VTe₂. (a) PDOS for V3d (d_{xy} , d_{yz} , d_{zx}). (b) PDOS for Te5p (p_x , p_y , p_z). (c) Schematic drawing of p_z orbital along with the d_{xy} σ -bonding.

3. Transport data fig. S2.b Are these data necessary? I found them a bit confusing when compared with the rich ARPES dataset. For example, looking at $x=0.1$ transport curve, the anomaly occurs at ~ 400 K while from the ARPES data (fig. 2, fig. 3 and supplementary section S5) it is stated that the CDW transition starts at ~ 250 K as stated in the main text. Can the authors comment a bit more on this?

Since the $V_{1-x}Ti_xTe_2$ compound and its phase diagram has not been reported so far, we think that the basic properties such as electrical resistivity and XRD should be helpful for readers. On the other hand, we admit that the data presentation had been confusing and partly misleading, since our definition of x for resistivity and XRD had been the nominal V:Ti ratio of the starting raw materials, not the grown sample. We find that the amount of Ti in polycrystals as well as CVT-grown single crystals fluctuates with some tendency toward deficiency. Since we precisely evaluated the Ti concentration for all the ARPES samples by using EDX, we now modify the electronic phase diagram by accounting on the ARPES results. Besides, to confirm the correspondence with ARPES, we newly measured the resistivity of single-crystalline $V_{0.90}Ti_{0.10}Te_2$ whose composition was determined by EDX. The resistivity anomaly shows up around 300 K (Fig. R6), which agrees well with the temperature-dependent ARPES.

To make these points clearer, we have modified Supplementary Note 2, by distinguishing the nominal starting composition x_n and the EDX-evaluated composition x . We have also added the new resistivity data (Fig. R6) in Supplementary Note, and corrected the caption of Fig. 1f as follows.

“f, Schematic electronic phase diagram for $V_{1-x}Ti_xTe_2$, based on the resistivity measurements.”

→

“f, Schematic electronic phase diagram for $V_{1-x}Ti_xTe_2$, based on the temperature-dependent ARPES measurements.”

Fig. R6: Electrical resistivity for single crystalline $V_{0.90}Ti_{0.10}Te_2$. The anomaly corresponding to the CDW transition is discerned at around 300 K.

Another point not completely clear to me is the apparent discrepancy between transport and ARPES data in the CDW phase: ARPES spectra show strong localisation of the electronic bands (across the M_2 high symmetry point) and the arise of almost 1D electronic state, while the transport shows metallic behaviour with localisation only for very low temperature ($<20K$). Can the authors clarify this point?

We thank the Reviewer for the comment. The Reviewer is correct that the bands around the \bar{M}_2 points indeed turn into the localized flat bands, accompanying the disappearance of corresponding Fermi surfaces. We think that the abrupt increase of resistivity on cooling below T_s (Fig. R6) should be reflecting this modification of the Fermi surface. However, there remain the Fermi surfaces at around \bar{M}_1 with well-defined k_z dispersion, and also around Γ points. We speculate that the carriers on these highly warped Fermi surfaces are responsible for keeping the conductivity until low temperature.

4. Spin-ARPES data fig. S8.d As the signal is faint, I would suggest to avoid the arb. units and to indicate the counts of the $I_{y+,-}$ spectra.

We thank the reviewer for his/her suggestion. Figures R7 (b) and (c) show the raw $I_y^{+(-)}$ spectra and their subtraction ($I_y^+ - I_y^-$) respectively at $\theta = \pm 4^\circ$. We can confirm the slight plus/minus contrasts near the Fermi level even in these raw data. In this revision, we have moved the spin-ARPES spectra $I_y^{\uparrow(\downarrow)}$ from previous Fig. S8 to the main text, according to the Reviewer #3's suggestion. At the same time, we now show the spin-ARPES geometry [Fig. R7 (a)]

and raw $I_y^{+(-)}$ spectra [Figs. R7 (b), (c)] in the new Supplementary Figure S8, to account for this Reviewer #2's comment.

Fig. R7: Experimental setup of spin-resolved ARPES. (a) Spin-resolved ARPES geometry. (b) The raw spectra ($I_y^{+(-)}$) and their subtraction ($I_y^+ - I_y^-$) at $\theta = -4^\circ$. (c) Same as (b), but at $\theta = +4^\circ$.

A curiosity on the transport data: I am wondering why the anomaly is fainter for higher x values. Is it due to disorder introduced by the doping or it depends on the hole concentration introduced by the Ti-doping?

Since the RRR (residual resistivity ratio) gets quickly suppressed by Ti doping (see Fig. S2), the disorder effect could be playing some role on smearing the anomaly. At the same time, maybe more importantly, the CDW itself gets suppressed on Ti doping as seen in the lowering of T_s , mainly due to the change of hole concentration. It could also suppress the T_s anomaly. To more quantitatively discuss the CDW amplitude and how it gets suppressed on Ti doping, the systematic diffraction measurements should be performed in future.

Reviewer #3 (Remarks to the Author):

The manuscript by Mitsubishi et al. reports changes in topological surface states in doped VTe₂, due to large changes in the crystal structure due to the formation of a 3x1 CDW. In principle, this is quite an interesting finding, although the way the paper is initially presented, it seems to imply (incorrectly) that the CDW transitions completely changes the topological character of the material which is not the case. Instead, some of the DSSes are removed due to CDW formation, while other DSSes remain present (in the ungapped parts of the Fermi surface). This is still an interesting finding, although not as remarkable as if the entire topological character of the material changed.

We thank Reviewer #3 for his/her invaluable time and effort on reviewing our manuscript. We are also happy to see the reviewer's comments pointing out the significance of our work.

Regarding the CDW transition and its influence on the entire topological character of this material, we admit that we cannot conclusively discuss it since the precise determination of whole electronic bands in the CDW phase is still lacking (will be discussed later in this reply). To cope with this issue, we changed the abstract and introduction part accordingly.

“... Spin- and angle-resolved photoemission spectroscopy with first-principles calculations reveal the huge anisotropic modification of the bulk electronic structure by the CDW formation, accompanying the **selective** disappearance of certain Dirac-type spin-polarized topological surface states that exist in the normal state. Thorough three dimensional investigation of bulk states indicates that the **corresponding** band inversion at the Brillouin zone boundary dissolves upon CDW formation, by transforming into anomalous flat bands. ...”

The presentation of the data showing the DSSes themselves is not particularly clear, particularly in Figures 4g-h, and Figures 3g-h. I actually found that some of the data in the SI (Figs S5 and S7) seem to show the DSSes much more clearly. I do believe that the authors' main identification of the DSSes in both the 1T and 1T' phases.

We thank the reviewer for his/her comment. We are glad to know that the reviewer appreciates our identification of the DSS. The data in previous Figs. 4g-h and Figs. 3g-h were recorded by using the photon energies of 54-69 eV to demonstrate the k_z -dependence, which is important for explaining the band inversion. There the DSS tends to lose its intensity possibly due to the matrix element effect, which is in principle inevitable. With the data in SI, we think we can convince the readers, as the reviewer has also evaluated.

In addition, there is some discussion of the spin-resolved results even in the abstract, yet none of the spin-resolved measurements actually show up in the main manuscript, only the supplemental Figure S8; authors should consider moving this to the main text. Granted, the spin-resolved data is not of the best quality, although the results from the lower branch of the DSS seem relatively clear.

We thank the reviewer for his/her valuable suggestion. We agree that it is preferable to present the spin-resolved

ARPES results in the main text. We have now moved the spin-resolved ARPES results (Fig. R8) and the relevant discussion to the main text. We have also reorganized the manuscript accordingly.

Fig. R8: Spin-resolved ARPES measurement on $1T''$ -VTe₂. (a) Schematic 2D BZ with mixed in-plane CDW domains. The red arrow qualitatively indicates the momentum cut of spin-resolved ARPES measurement. (b) ARPES image with the peak plots of spin-resolved spectra (s polarized 21 eV photons, 15 K). The domains are not separated. Red (blue) triangle markers indicate the peak positions of spin-up (-down) spectra in \bar{M}_1 side, whereas purple (cyan) for \bar{M}_2 side. (c) Spin-resolved spectra around \bar{M}_1/\bar{M}_2 points. (d) Spin-resolved spectra near the Fermi level at $\theta = \pm 4^\circ$.

Furthermore, the DFT calculations in Figure 3a-c in the normal state structure appear to indicate band inversion, supported by the appearance of a DSS in the slab calculation in 3c. This seems to be sensible and consistent with the ARPES data in the normal state. What about the DFT calculations for a 3×1 supercell in the $1T'$ phase? Do they likewise support the author's observations of a DSS at certain k-points and absence on the gapped parts of the FS? I see in the SI that they have produced DFT calculations of the $1T'$ phase but am surprised that they have not produced a similar analysis.

We thank the reviewer for his/her valuable comment. We fully agree that the slab calculation on $1T''$ phase should be very helpful for the present discussion on topological properties. Nevertheless, since the $1T''$ phase has a complicated crystal structure with many atoms in the whole unit cell (the number of atoms in $1T''$ is 9 times of $1T$), its slab calculation is technically demanding. Actually, after receiving this comment, we re-noticed its importance and had been trying the slab calculation in the similar manner with the normal $1T$ case until recently. Regrettably, however, it ended up with an unsuccessful attempt (did not converge). One problem is the calculation cost itself, but even if it is overcome, we still have certain difficulty in unfolding the obtained slab bands into the $1T$ BZ, to compare with ARPES. We have to wait for the future progress in the calculation techniques and/or power.

Besides, we also note that the bulk calculation for $1T''$ -VTe₂ itself is still lacking the quantitative accuracy for discussing the entire topological character. Though the qualitative parts of the ARPES results (e.g. the drastic change of Fermi surface and the flat band formation) are quite well reproduced, the energies of the flat bands for example are still in disagreement: They are located at binding energies of ~ 0.25 eV (ARPES) and ~ 0.4 eV (DFT).

Since the topological character sharply depends on the relative energies (crossings and anti-crossings) of the multiple bands, they must be precisely reproduced in the calculation before discussing the realistic topological classification. Here we note that VTe_2 does not appear in the recently developed topological materials databases (<https://www.topologicalquantumchemistry.com/>, <http://materiae.iphy.ac.cn/>, started to operate from early 2019) as well, possibly due to the complex structure as mentioned above. In this viewpoint, we would rather like to stress that our present experimental results and the microscopic discussion offer an important key for exploring the topological matters which are still too complex to predict in the present stage.

To clearly state this situation, we added some description as the following.

(On page 12, line 13)

Finally, we would like to mention that the topological character of materials should be determined by considering the detailed band structures for the whole system and quantifying the Berry curvature and topological invariants (Z_2 index, (spin) Chern number, etc). The difficulty in the present case, however, is that the accuracy of the band calculation on $1T''$ phase is still insufficient. Though the qualitative features (such as the quasi-one-dimensional Fermi surface and the flat band formation) are quite well reproduced (see Supplementary Note 12 for the band unfolding calculation), the detailed band structures including their relative energy positions are not in good enough agreement. This makes it difficult to fully discuss the topology which sharply depends on the crossings of band energy levels, particularly on its modification across the CDW transition. In this viewpoint, we would rather like to stress that our present results and microscopic analysis are offering one way to pursue the topological character of a complex material that is still difficult to predict.

Also (a more minor question), why is a Ti doping of 13% selected (as it does not appear to correspond to any of the doping used), nor is a supercell used in the calculation?

We thank the reviewer for his/her comment. In this manuscript, we presented the ARPES data of $1T$ phase taken for both Ti 10%-doped (Figs. 2c, 3d, 3g-j of the previous manuscript) and Ti 13%-doped (Figs. 3e, 5d) samples. Actually, these samples were picked up from an identical batch, however, we found variations of the CDW transition temperatures (~ 280 K vs. ~ 250 K) by ARPES. To confirm, we conducted EDX measurement afterwards and found that the Ti concentration are fluctuating even in the same batch. In this way, the data of 10% and 13% samples got mixed unintentionally. Nevertheless, here we would like to stress that the difference of Ti13% and Ti10% is very small beside the transition temperatures, and do not affect our conclusions.

For the band calculation, we adopted the virtual crystal approximation (VCA), not the supercell method, since the VCA enables us the calculation for arbitrary $\text{V}_{1-x}\text{Ti}_x\text{Te}_2$ compositions, and also without changing the symmetry. Since Ti and V are the adjacent elements, VCA is expected to work well enough for $\text{V}_{1-x}\text{Ti}_x\text{Te}_2$ system. Doping of 13% was selected simply just to compare with one of the ARPES result.

The organization and presentation of the paper is jumbled, disorganized, and confusing. There is a lot of switching back and forth between different data sets (single domain, domain-averaged, different photon energies, etc...). It

would be much more clear-cut to just have all the data taken for single domain, since the domains appear to be rather large (many hundreds of microns – this is not a technically challenging experiment). As mentioned earlier, it is not evident until quite late in the manuscript (end of pg 7, beginning of pg 8) that only some DSSes disappear and others remain unchanged – this should be made clear at the outset.

We thank the Reviewer #3 for his/her valuable comment. We agree that it would be ideal to collect all the data by using the single-domain samples. However, in reality, it is difficult to accomplish because of following reasons: (1) The domain size attains several hundreds of microns in some best cases, but in many other cases, it is much smaller (<100 micron or much less). (2) Especially for spin-resolved ARPES, the beam spot size is around 1mm, which is large compared to the domain size. (3) Since the synchrotron beam-time is severely limited, it is not easy to always find the right big-enough single domain and perform the thorough (k_z , T -dependent) measurement. We hope the Reviewer will understand our situation. Taking his/her suggestion seriously, however, we have reconsidered the overall structure of the paper so that the data of the single-domain should appear in the early stage. We have revised the whole manuscript as follows.

- The abstract has been modified to more unambiguously explain the selective disappearance of the DSSes.
- “Results and Discussions” now starts with a brief overview of electronic structure for $1T$ and $1T''$ phases by demonstrating the single-domain ARPES (Fig. R9). With this, we believe it has become much easier for readers to interpret the following results on domain-mixed samples.
- We have explicitly denoted as “multi-domain” in figures collected with CDW domain-mixed samples.
- We have consistently used the $1T''$ notation (\bar{M}_1/\bar{M}_2 , \bar{K}_1/\bar{K}_2) both for single- and multi-domain $1T''$ data (In the previous manuscript the $1T$ and $1T''$ notations were mixed for multi-domain data).
- We revised the text in Supplementary Note 5 to describe the realistic domain issue.

We believe that these revisions are very helpful for readers.

Fig. R9: Overview of Fermi surface and band dispersions for normal $1T$ and CDW $1T''$ phases. (a) Bird’s eye ARPES image of $1T$ - $V_{0.87}Ti_{0.13}Te_2$ (300 K, $h\nu = 21.2$ eV). (b) Same as a, but for $1T''$ - VTe_2 (200 K, $h\nu = 90$ eV).

Summary of Changes

The changes are also **highlighted (with yellow)** in the revised manuscript except for ...

- renumbering of original (unchanged) figures/references/Supplementary Notes.
- replacement of “Section S#” with “Supplementary note #”.

Main text:

- **page 1:** We added the coauthors (H. Takahashi and H. Sakai) who contribute our work.
- **page 2, line 9:** We modified the words “the disappearance of certain Dirac-type ...” into “the selective disappearance of Dirac-type ...”.
- **page 2, line 11:** We added the word “corresponding”.
- **page 3, line 23:** We added the sentence “and purple bronze AMo_6O_{17} ($A = Na, K$)²⁵⁻²⁷” and the references #25-27.

[25] M.-H. Whangbo, E. Canadell, P. Foury, J.-P. P. Hidden Fermi Surface Nesting and Charge Density Wave Instability in Low-Dimensional Metals. *Science*. **252**, 96–98 (1991).

[26] Mou, D. *et al.* Discovery of an Unconventional Charge Density Wave at the Surface of $K_{0.9}Mo_6O_{17}$. *Phys. Rev. Lett.* **116**, 196401 (2016).

[27] Su, L., Hsu, C. H., Lin, H. & Pereira, V. M. Charge Density Waves and the Hidden Nesting of Purple Bronze $K_{0.9}Mo_6O_{17}$. *Phys. Rev. Lett.* **118**, 257601 (2017).

- **page 4, line 12:** We moved the sentences of paragraph in line 23, page 4 of the previous manuscript (“Figure 1f displays the schematic electronic phase diagram of ...”) here, and modified the words “based on the temperature-dependent resistivity measurements (see Section S2)” into “based on the temperature-dependent ARPES measurements” in line 13, page 4.
- **page 4, line 23:** We modified the sentence “the details of BZ are shown in Section S1, Supplementary information” into “see Supplementary Note 1 for the details of BZ”.
- **page 5, line 2:** We added the new paragraph as follows.

Let us start by briefly overviewing the anisotropic modification of electronic structures from $1T$ to $1T'$ by presenting the ARPES data successfully focused on a CDW single-domain region. Figure 2a shows the ARPES image for the normal state $1T-V_{0.87}Ti_{0.13}Te_2$ taken at 300 K with a He discharge lamp (photon energy $h\nu = 21.2$ eV). We find V-shaped band dispersions along $\bar{K} - \bar{M} - \bar{K}$ that clearly cross the Fermi level (E_F). Looking at the higher binding energy (E_B) region, these V-shaped bands are connected to the Dirac-cone-like bands reminiscent of surface states in topological insulators, with the band crossing (Dirac points) at \bar{M} . The topological character of these Dirac bands will be discussed later. On the other hand, Fig. 2b displays the ARPES results on a single-domain CDW state in $1T'-VTe_2$ (200 K, synchrotron light $h\nu = 90$ eV). Because of the zigzag type CDW formation, the system now loses the 3-fold rotational symmetry, and the 3 equivalent \bar{M} points in $1T$ turn into one \bar{M}_1 and two \bar{M}_2 . Here, the V-shaped band and Dirac-like bands remain at \bar{M}_1 , whereas at \bar{M}_2 side the unusual flat band is observed and the Dirac-like state is vanished. Thus, the $1T-1T'$ CDW transition induces the huge directional change of electronic structure accompanying the selective

disappearance of Dirac-like states. In the following, we discuss these band structures in detail, by comparing with band calculations.

- **page 5, line 19:** We modified the sentence "It is characterized ... around $\bar{\Gamma}$ and \bar{K} , respectively" into "They are characterized ... respectively around Γ and K ".
- **page 5, line 21:** We removed the words "photon energy".
- **page 5, line 25:** We added the sentence "(We note that the tiny electron pocket at Γ is not clearly detected, probably reflecting the finite energy mismatch of the bands compared to the calculation.)"
- **page 6, line 2:** We removed the word " (E_F) ".
- **page 6, line 3:** We changed the sentence "To grasp the essential electronic modification *via* the CDW formation, we investigate low temperature $V_{0.90}Ti_{0.10}Te_2$ and VTe_2 ." (page 5, line 14 of the previous manuscript) as follows.
To grasp the essential electronic modification *via* the CDW formation, we survey the temperature- and doping-dependent ARPES results (He discharge lamp, $h\nu = 21.2$ eV). Note that the $1T$ phase inevitably contains the in-plane 120-degree CDW domains reflecting the 3-fold symmetry of $1T$, and usually ARPES measurements include the signals of multiple domains (see Supplementary Note 5).
- **page 6, line 8:** We added the word "multi-domain". Similar change is done for line 14 of page 6 and Figs. 3eg, 5j-l.
- **page 6, line 8:** We added the word "(#1)".
- **page 6, line 9:** We removed the word " (E_B) ".
- **page 6, line 11:** We changed the word " \bar{K} " into " \bar{K}_1/\bar{K}_2 ". Similar change is done for line 11 of page 9,
- **page 6, line 13:** We changed the sentence "We note that the coexistence of these flat/dispersive bands results from the mixing of in-plane 120-degree CDW domains (see Section S6), as discussed later." (page 5, line 21 of the previous manuscript) into "Such coexistence of flat/dispersive bands results from the mixing of CDW domains."
- **page 6, line 22:** We change the word "start by introducing" into "introduce".
- **page 6, line 24:** We change the sentence "Figure 3a shows the calculations of $1T-V_{0.87}Ti_{0.13}Te_2$ at several k_z , plotted along the momentum parallel to Γ -M (*i.e.* k_{TM})." (page 6, line 8 of the previous manuscript) into "The calculations of $1T-V_{0.87}Ti_{0.13}Te_2$ at several k_z , plotted along the momentum parallel to Γ -M (k_{TM} , see Fig. 4a) are displayed in Fig. 4b."
- **page 7, line 1:** We modified the word "shown" into "depicted".
- **page 7, line 1:** We added the words "(see Supplementary Note 3 for detailed orbital components)".
- **page 7, line 3:** We added the words "blue-like" and "red-like".
- **page 8, line 16:** We removed the sentence "Our spin-resolved ARPES measurement reveals that this Dirac band has the spin-polarization similar to the case of surface states in topological insulators (see Section S8), thus indicating its topological nature." (page 7, line 24 of the previous manuscript).
- **page 8, line 17:** We modified the sentence "Thus, the CDW induces the huge directional change of electronic structure accompanying the disappearance of certain Dirac surface states." (page 8, line 3 of the previous manuscript) into "These indicate that the CDW induces the drastic directional modification of electronic structure."

- **page 8, line 20:** We added the new paragraph as follows.

Here, to confirm the topological nature of the Dirac surface state in VTe₂, we performed spin-resolved ARPES on a multi-domain sample (see Supplementary Note 8 for the experimental setup). Figure 5f depicts the schematic 2D BZ with CDW multi-domains, together with the measurement region (the red arrow). Figure 5g shows the spin-integrated ARPES image with the peak plots obtained from spin-resolved spectra (15 K, $h\nu = 21$ eV, s polarization). By comparing with the results from the single-domain measurement (Figs. 5b-e), these peaks can be easily assigned to either \bar{M}_1 or \bar{M}_2 side. The red (blue) and purple (cyan) triangle markers respectively represent the peak positions of spin-up (-down) spectra at \bar{M}_1 and \bar{M}_2 sides. As shown in the spin-resolved spectra in Fig. 5h, the lower branch of the Dirac cone band clearly shows the spin polarization with sign reversal at \bar{M}_1 point (*i.e.* Dirac crossing point), similarly to the topological surface state in the topological insulators. On the other hand, the bulk flat bands around $E_B \sim 0.25$ eV at \bar{M}_2 side indeed show the spin degenerate character. Figure 5i is an expanded viewgraph of spin-resolved spectra near the Fermi level at emission angle $\theta = \pm 4^\circ$. There are slight spin-up/down intensity contrasts just below E_F ($E_B \sim 0.05$ eV), which should be corresponding to the upper branch of the surface Dirac cone band at \bar{M}_1 side.

- **page 9, line 12:** We changed the sentence “By comparing with the results from the domain-selective measurement (Figs. 4b-e), these peaks can be easily assigned to either \bar{M}_1 or \bar{M}_2 side (see Section S9).” into “Again by comparing with the single-domain measurement, we can assign the peaks to either \bar{M}_1 or \bar{M}_2 side (see Supplementary Note 9).”
- **page 9, line 24:** We changed the sentence (page 8, line 20 of the previous manuscript) as follows.

For simplicity, we focus on $V3d t_{2g}$ orbitals that dominate the density of states near E_F (see Section S10). We adopt an orthogonal XYZ coordination by considering the VTe₆ octahedron as shown in Fig. 5a.

↓

For simplicity, we adopt an orthogonal octahedral XYZ coordination by considering the VTe₆ octahedron as shown in Fig. 6a, and focus on $V3d t_{2g}$ (d_{XY} , d_{YZ} , d_{ZX}) orbitals that dominate the density of states near E_F (see Supplementary Note 10).

- **page 10, line 4:** We changed the words “the b_m axis shown in Fig. 1d” into “the vanadium’s chain direction b_m ”.
- **page 10, line 14:** We changed the words “Supplementary Note 10 and 11” into “Supplementary Note 10”.
- **page 10, line 15:** We changed the word “Figs. 2e, 2g, and 4d” into “Fig. 5d”.
- **page 10, line 23:** We added the words “and purple bronzes AMo_6O_{17} ($A = Na, K$)^{25–27}”.
- **page 11, line 5:** We removed the word “Finally,”.
- **page 11, line 21:** We added the new paragraphs as follows.

Here we attempt an argument on the band inversion in association with CDW based on the change in crystal symmetry. In the high-temperature $1T$ phase ($P\bar{3}m1$), as mentioned, the Dirac cones appear at the \bar{M} points, due to the band inversions involving the Te5 p and V3 d orbitals occurring along M-L lines. Here, the M-L line resides in a mirror plane, which prohibits the hybridization between the band consisting of Te5 p anti-bonding orbitals (odd with respect to the mirror plane) and that derived from the predominantly V3 d with finite Te5 p bonding orbitals (even with respect to the mirror plane), without the help of SOC. Eventually, the non-

relativistic bands cross at $k_z \sim 0.76 \pi/c$ along M-L depicted as the broken curves in Fig. 4b, thus causing the band inversion, due to the k_z dependent energy eigenvalues. Only by the SOC, they get mixed and make a gap (~ 200 meV) around the crossing point. In the low-temperature $1T'$ phase ($C2/m$), on the other hand, while the \bar{M}_1 point keeps the similar situation with the \bar{M} point in high- T phase, \bar{M}_2 no longer resides in the mirror plane. At \bar{M}_2 , the $V3d$ and $Te5p$ (bonding, antibonding) can now significantly mix and form the well hybridized flat band with no k_z dependence, leading to the disappearance of the band inversion. This can be also confirmed by evaluating the orbital components in our bulk calculation (see Supplementary note 11). It indicates the significant d_{YZ}/d_{ZX} - p_Z hybridization (here X, Y, Z represents the octahedral setting, see Fig. 6a) forming the flat band at the binding energy of ~ 0.4 eV. In contrast, the mirror plane remains for $\bar{\Gamma} - \bar{M}_1$, thus the band inversion and the related Dirac surface state can sustain.

Finally, we would like to mention that the topological character of materials should be determined by considering the detailed band structures for the whole system and quantifying the Berry curvature and topological invariants (Z_2 index, (spin) Chern number, etc). The difficulty in the present case, however, is that the accuracy of the band calculation on $1T'$ phase is still insufficient. Though the qualitative features (such as the quasi-one-dimensional Fermi surface and the flat band formation) are quite well reproduced (see Supplementary Note 12 for the band unfolding calculation), the detailed band structures including their relative energy positions are not in good enough agreement. This makes it difficult to fully discuss the topology which sharply depends on the crossings of band energy levels, particularly on its modification across the CDW transition. In this viewpoint, we would rather like to stress that our present results and microscopic analysis are offering one way to pursue the topological character of a complex material that is still difficult to predict.

- **page 15, line 2:** We added the paragraph as follows.

Spin-resolved ARPES measurements were performed at Efficient SPin Resolved SpectroScOpy (ESPRESSO) end station attached to the APPLE-II-type variable polarization undulator beamline (BL-9B) at the Hiroshima Synchrotron Radiation Center (HSRC)³⁶. The analyzer of this system consists of two sets of very-low-energy electron diffraction (VLEED) spin detectors, combined with a hemispherical electron analyzer (VG-Scienta R4000). Measurements of $1T'$ -VTe₂ were performed using a s -polarized 21 eV light at 15 K. In the spin resolved measurements, we set the energy and angle resolutions to 120 meV and ± 1.5 deg, respectively. We adopted the Sherman function $S_{\text{eff}} = 0.25$ for analyzing the obtained data. The data in Figs. 5g-i were obtained with this condition.

- **page 15, line 4:** We added the reference #36.

[36] Okuda, T. *et al.* Efficient spin resolved spectroscopy observation machine at Hiroshima Synchrotron Radiation Center. *Rev. Sci. Instrum.* **82**, 103302 (2011).

- **page 15, line 18:** We added the reference #38.

[38] Bellaiche, L. & Vanderbilt, D. Virtual crystal approximation revisited: Application to dielectric and piezoelectric properties of perovskites. *Phys. Rev. B - Condens. Matter Mater. Phys.* **61**, 7877–7882 (2000).

- **page 16, line 1:** We change the sentence “We adopted the reported crystalline structures of $1T$ - and $1T'$ -VTe₂^{20,36}” into “We adopted and fixed the crystalline structures of $1T$ - and $1T'$ -VTe₂ reported in ref. ^{20,41}”.

- **page 16, line 4:** We added the statement about *Data Availability*.
- **page 20, line 2:** We removed the word “H. Takahashi”.
- **page 20, line 11:** We added the words “H.T., H.S.”.
- **Fig. 1a:** The *xyz* axes are added.
- **Fig. 1f:** We changed the words in caption “resistivity measurements” into “temperature-dependent ARPES measurements”.
- **Fig. 2:** We added a new Fig. 2 as follows.

Figure 2 | Overview of Fermi surface and band dispersions for normal 1T and CDW 1T' phases. a, Bird's eye ARPES image of 1T- $V_{0.87}Ti_{0.13}Te_2$ (300 K, $h\nu = 21.2$ eV). **b,** Same as a, but for 1T'- VTe_2 (200 K, $h\nu = 90$ eV).

- **Fig. 3:** We changed the title “Overview of the electronic structures in the normal (1T) and CDW (1T') phases.” into “Temperature/Ti-doping dependence of electronic structures.”
- **Fig. 3a:** The lines are thickened.
- **Fig. 3b:** We replaced the schematic FS with the calculational FS and modified the caption.
- **Figs. 3e-h:** We changed “ \bar{M} (\bar{K})” into “ \bar{M}_1/\bar{M}_2 (\bar{K}_1/\bar{K}_2)”.
- **Fig. 4:** We changed the title “Topological surface state in the normal 1T phase for Ti-doped VTe_2 .” into “Topological character of band structures in the normal 1T phase for Ti-doped VTe_2 .”
- **Fig. 4a:** We added a new figure and caption as Fig. 4a as follows.

Figure 4 | Topological character of band structures in the normal 1T phase for Ti-doped VTe_2 . **a,** The Brillouin zone of 1T phase. k_x, k_y, k_z indicate the axes in the reciprocal space corresponding with the global *xyz* axes (see Fig. 1a) used for the calculation in **b**.

- **Fig. 4b:** We enlarged the letters in figure and corrected the wrong capitalization in caption (The Bulk → The

bulk).

- **Fig. 5f:** We added the notations “CDW (0 0 1) BZ (multi-domain)” and “ \bar{M}_1/\bar{M}_2 ” in figure and modified the caption.
- **Figs. 5g-i:** We added new figures and captions as Figs. 4g-i as follows.

Figure 5 | Anisotropic electronic structures in the CDW 1T' phase for VTe₂. **f**, Schematic 2D BZ with mixed in-plane CDW domains. The red arrow qualitatively follows the measurement direction of spin-resolved and $h\nu$ -dependent ARPES. **g**, ARPES image with the peak plots of spin-resolved spectra (s polarized 21 eV photons, 15 K). The domains are not separated. Red (blue) triangle markers indicate the peak positions of spin-up (-down) spectra of \bar{M}_1 side bands, whereas purple (cyan) for \bar{M}_2 side. **h**, Spin-resolved spectra around \bar{M}_1/\bar{M}_2 points. **i**, Spin-resolved spectra near the Fermi level at $\theta = \pm 4^\circ$.

- **Fig. 6a:** We added the xyz axes.
- **Fig. 6c:** We added the mirror plains in figure and the illustration in caption.
- **Fig. 6d:** We replaced “ \bar{M} ” with “ \bar{M} ” and added the notations of “ \bar{K} (\bar{K}_1, \bar{K}_2)” and color scale bars in figure.

Supplementary information

- **Figs 1b,d:** We added the mirror plain(s) in figures and the illustrations in captions.
- **page 3, line 1:** We removed the word “polycrystalline”.
- **page 3, line 2:** We added the words “To grasp the overall phase diagram of $V_{1-x}Ti_xTe_2$,” and removed the words “of $V_{1-x}Ti_xTe_2$ ” (page 3, line 2 of the previous manuscript).
- **page 3, line 4:** We changed the words “nominally, $1-x : x : 2$ ” into “ $1 - x_n : x_n : 2$ ”.
- **page 3, line 7:** We added the words “of $V_{1-x_n}Ti_{x_n}Te_2$ ” and the sentence “Here we note that x_n , the nominal ratio of the raw material powders, does not precisely correspond to the actual x , due to some fluctuation of composition occurring during the growth.”.
- **page 3, line 9:** We changed the words “For $x = 0$ and 0.10,” into “For lower x_n compositions,”.
- **page 3, line 9:** We removed the words “($x \geq 0.15$)” (page 3, line 9 of the previous manuscript).
- **page 3, line 16:** We changed the word “ $x \geq 0.3$ ” into “ $x_n \geq 0.3$ ”.
- **page 3, line 18:** We added the sentence “These results suggest that the 1T-1T' transition temperature in this system is favorably controllable by tuning the Ti doping level.”
- **page 3, line 20:** We changed the sentences (page 3, line 16 of the previous manuscript) as follows.

Since the CVT-grown single crystals may contain some amount of composition fluctuation, the Ti contents (x) for respective samples were determined by the EDX measurement. From temperature-dependent ARPES measurements, the transition temperatures of $x = 0.10$ and 0.13 single crystals were respectively estimated to be around 280 K and 250 K (see also Section S5).

↓

Here, the Ti contents (x) for respective samples were precisely determined by the EDX measurement, to perform the systematic x -dependent study. Figure S2c shows the electrical resistivity for single crystalline $V_{0.90}Ti_{0.10}Te_2$ ($x = 0.10$). The anomaly corresponding to the CDW transition is discerned at around 300 K, which agrees with the temperature-dependent ARPES measurements shown in Supplementary Note 6.

- **Fig. S2:** we removed the word “polycrystalline” from the title.
- **Fig. S2a:** We replaced “ x ” with “ x_n ” and added “poly.” in figure, and changed the caption as follows.
Here x is the nominal concentration. Miller indices based on the space group $C2/m$ ($0 \leq x \leq 0.1$) and $P\bar{3}m1$ ($0.15 \leq x \leq 0.5$) are depicted.
- ↓
Here x_n is the nominal concentration of raw material powders and not precisely same as the actual x value of the grown samples. Miller indices based on the space group $C2/m$ ($\bar{6}03$ and 301) and $P\bar{3}m1$ (101) are depicted.
- **Fig. S2b:** We replaced “ x ” with “ x_n ” and added “poly.” in figure.
- **Fig. S2c:** We added a new figure S2c as follows.

Fig. S2c: Electrical resistivity for single crystalline $V_{0.90}Ti_{0.10}Te_2$ ($x = 0.10$). The anomaly corresponding to the CDW transition is discerned at around 300 K.

- **page 5, line 3:** We changed the sentences (page 5, line 3 of the previous manuscript) as follows.
Figures S3a and S3b show the band dispersions along the high symmetry lines and the resulting Fermi surface, respectively. Our ARPES data taken with a He discharge lamp ($h\nu = 21.2$ eV, Figs. 2c and 2d in the main text) qualitatively agree well with the calculation in the $k_z = 0$ (Γ -K-M- Γ) plane.
- ↓
Figures S3a displays the resulting 3D Fermi surface. Figures S3b-f show orbital-weighted band calculation in the global xyz setting (see also Fig. 4b in the main text).
- **Fig. S3a:** We changed the caption “The corresponding Fermi surface” into “3D view of Fermi surface”.
- **Figs. S3b-f:** We removed the all-orbital-projected band calculation (Fig. S3a of the previous manuscript).

Instead, we added five orbital-separated band calculations as follows.

Figure S3 | Calculated electronic structure of 1T-V_{0.87}Ti_{0.13}Te₂. **a**, 3D view of Fermi surface. **b-f**, Orbital-weighted band dispersions for p_x+p_y (**b**), p_z (**c**), $d_{x^2+y^2}+d_{xy}$ (**d**), d_{z^2} (**e**), and $d_{xz}+d_{yz}$ (**f**).

- **Fig. S4:** We changed the notations of 1T' phase.
- **page 7, line 3:** We changed the words “a typical” into “one examples of the”.
- **page 7, line 4:** We added the words “single crystal with”.
- **page 7, line 5:** We changed the words “can be confirmed. They reflect” with “confirmed here reflect”.
- **page 7, line 6:** We changed the sentences (page8, line 6 of the previous manuscript) as follows.
 On the other hand, the orientation of the stripe patterns rotates by 120 degrees in a longer length-scale, typically of 600-1000 μm . From this, ...
 ↓
 On the other hand, in some cases, the orientation of the stripe patterns rotating by 120 degrees is observed in the length-scale of 10-1000 μm , as seen in Fig. S5a. From these, ...
- **page 7, line 19:** We change the word “the” into “a”.
- **page 8, line 6:** We added the sentence “Thus, here we simply use 1T notation instead of 1T' (e.g. \bar{M} instead of \bar{M}_1/\bar{M}_2).”
- **page 10, line 1:** We changed the Supplementary Note title “Spin-resolved ARPES measurement” into “Setup of spin-resolved ARPES measurement”.
- **page 10, line 2:** We removed the first paragraph in the previous manuscript (“Spin-resolved ARPES measurements were performed at ...”) to Method in the main text. Instead, we began with the sentence as follows.
 Figure S8a shows the geometry of spin-resolved ARPES measurements for multi-domain VTe₂ (see also Methods in the main text). To access the $\bar{K}_1 - \bar{M}_1 - \bar{K}_1/\bar{K}_1 - \bar{M}_2 - \bar{K}_2$ lines, the attached samples are ...
- **page 10, line 19:** We also modified the following statement (“Figure S8b shows ...” of the previous manuscript) as follows.

The spin-resolved ARPES data in the main text (Figs. 5g-i) are collected with *s* polarized 21 eV light at $\varphi = 28^\circ$ and $-4^\circ \leq \theta \leq +4^\circ$ setting. The sample orientation is arranged so that the sample axes *X* and *Y* become parallel to the \mathbf{a}_m and \mathbf{b}_m crystal axes for one of the 3 in-plane multi-domain. Figures S8b and c show the raw $I_y^{+(-)}$ spectra and their subtraction ($I_y^+ - I_y^-$) respectively at $\theta = \pm 4^\circ$ (Note that after conversion $I_y^{\uparrow(\downarrow)}$ is displayed in Fig. 5i in the main text). We can find the slight plus/minus contrasts near the Fermi level in these raw data.

- **Figure S8:** We modified Fig. S8 as follows.

Figure S8 | Experimental setup of spin-resolved ARPES. **a**, Schematic spin-resolved ARPES geometry. The attached samples are rotated around the polar (θ) and tilt (φ) axes to measure the band dispersion along $\bar{K}_1 - \bar{M}_1 - \bar{K}_1/\bar{K}_1 - \bar{M}_2 - \bar{K}_2$. **b**, The raw $I_y^{+(-)}$ spectra and their subtraction ($I_y^+ - I_y^-$) at $\theta = -4^\circ$. **c**, Same as **b**, but at $\theta = +4^\circ$.

- **page 11, line 10:** We changed the Supplementary Note title “ARPES images and EDCs/MDCs along $\bar{K} - \bar{M} - \bar{K}$ for $1T''$ -VTe₂.” into “ARPES images and EDCs/MDCs along $\bar{K}_1 - \bar{M}_1 - \bar{K}_1/\bar{K}_1 - \bar{M}_2 - \bar{K}_2$ for $1T''$ -VTe₂”.
- **page 11, line 11:** We changed the word “ $\bar{K} - \bar{M} - \bar{K}$ ” into “ $\bar{K}_1 - \bar{M}_1 - \bar{K}_1/\bar{K}_1 - \bar{M}_2 - \bar{K}_2$ ”.
- **page 13, line 6:** We added the statement “The color and size of circles respectively represent the *d* orbital component and its weight (Note that the small size of circle markers indicate the dominance of Te5*p*).”.
- **page 15:** We added a new Supplementary Note 11 as follows.

Supplementary Note 11: CDW effects on Te5*p* orbitals.

Figure S11a and b show the calculated PDOS of $1T''$ -VTe₂, respectively for V3*d* (d_{xy} , d_{yz} , d_{zx}) and Te5*p* (p_x , p_y , p_z). They indicate that the flat band at the binding energy of 0.3 ~ 0.4 eV is formed by the hybridization of d_{yz}/d_{zx} and p_z orbitals. Here we note that d_{yz} and d_{zx} well hybridize with p_z in the edge-sharing VTe₆ octahedral network, whereas the overlap for d_{xy} and p_z is negligibly small (see Fig. S11c).

Figure S11 | PDOS of 1T'-VTe₂. **a**, PDOS for V3d (d_{xy} , d_{yz} , d_{zx}). **b**, PDOS for Te5p (p_x , p_y , p_z). **c**, Schematic drawing of p_z orbital along with the d_{xy} σ -bonding.

- **page 16, line 8:** We added the word “single-domain”.
- **page 16, line 9:** We removed the words “show reasonable agreement and”.
- **Fig. S12d:** We added the word “single-domain” in caption.
- **page 18:** We moved the reference (Okuda, T. *et al. Rev. Sci. Instrum.* (2011).) from Supplementary Information to the main text.

REVIEWERS' COMMENTS:

Reviewer #1 (Remarks to the Author):

I have reviewed in detail the resubmitted manuscript, the reports of the other two reviewers, and the authors' responses to the reviewers' comments. The authors have made a genuine effort in clarifying in detail all the issues raised in those reports. They have significantly and extensively improved the manuscript by adding or updating figures, appropriately adding additional supplemental material, as well as by providing additional experimental details and discussion in the main text. It is now clear which results and conclusions are solid, and the limitations in addressing some of the outstanding questions (such as the DFT calculations for the ground-state of the 1T' phase, whose numerical difficulties I now better appreciate and understand).

With these changes, the overall set of results here has become even more compelling. The manuscript is now suitable for publication.

Reviewer #2 (Remarks to the Author):

The revised manuscript by Mitsuishi et al. maintains the high quality of data and analysis, and clarifies all the points raised in my review. I do think the paper in this form deserves publication in Nature Communications, with some minor modifications (see below)

In particular, the discussion of the accuracy of calculation, the clarification on the selective change of Dirac bands, and the unclear point regarding the composition vs. temperature of the transition have been clearly addressed in the present revised version. Accordingly, changes in the main text, figures and supplementary have been included.

I think that, still, two minor points should be addressed prior publication:

- a) regarding fig. R6 (S2c), authors should include, either in the text or in figure caption, their phrase regarding the highly warped Fermi surfaces (end of point 3 in their reply on page.12), this will further clarify their results
- b) authors may consider to slightly modify Figure 5 panel i). In fact, being panel i) extracted from h), and being the difference very small (as clearly seen in the corresponding figure in the supplementary), I would suggest to select e.g. only the 0.2 eV to -0.2 eV section to highlight the presence of a difference.

Reviewer #3 (Remarks to the Author):

I have reviewed the response by Mitsuishi et al. It is clear that the authors have invested a significant amount of effort to address my concerns, as well as those of the other reviewers. The authors have addressed or responded to my concerns regarding the presentation of their data, including the addition of spin-resolved results, the doping levels, and organization of the manuscript.

However, the remaining concern of mine is the most important one, and is also echoed in the comments of Reviewer #2 ("...the presented interpretation is not conclusive in that sense...as the topology of a material is a property of the system as a whole, the data seems to demonstrate that the topology is conserved across the CDW transition...this is "the" focal point/question of the paper and needs to be addressed more convincingly").

Unfortunately, it appears that (despite the authors' efforts) the DFT calculations are not able to answer this question conclusively, presumably due to the complex crystal structure. The authors make some arguments which are plausible, but not concrete. Without a clear resolution of this issue, which (in my opinion) is the key result of this paper, I am still reluctant to recommend this manuscript for publication.

Reviewer #1 (Remarks to the Author):

I have reviewed in detail the resubmitted manuscript, the reports of the other two reviewers, and the authors' responses to the reviewers' comments. The authors have made a genuine effort in clarifying in detail all the issues raised in those reports. They have significantly and extensively improved the manuscript by adding or updating figures, appropriately adding additional supplemental material, as well as by providing additional experimental details and discussion in the main text. It is now clear which results and conclusions are solid, and the limitations in addressing some of the outstanding questions (such as the DFT calculations for the ground-state of the 1T' phase, whose numerical difficulties I now better appreciate and understand).

With these changes, the overall set of results here has become even more compelling. The manuscript is now suitable for publication.

We sincerely thank again Reviewer #1 for his/her invaluable time and effort for reviewing our revised manuscript. We are very glad to see that he/she highly evaluates our work and recommends the publication in Nature Communications.

Reviewer #2 (Remarks to the Author):

The revised manuscript by Mitsuishi et al. maintains the high quality of data and analysis, and clarifies all the points raised in my review. I do think the paper in this form deserves publication in Nature Communications, with some minor modifications (see below)

In particular, the discussion of the accuracy of calculation, the clarification on the selective change of Dirac bands, and the unclear point regarding the composition vs. temperature of the transition have been clearly addressed in the present revised version. Accordingly, changes in the main text, figures and supplementary have been included.

We sincerely thank again Reviewer #2 for his/her invaluable time and effort for reviewing our revised manuscript. We are very happy to see that he/she highly evaluates our work and recommends the publication in Nature Communications. The reviewer pointed out several suggestions that are helpful for improving our manuscript. In the following, we answer all the comments point-by-point.

I think that, still, two minor points should be addressed prior publication:

a) regarding fig. R6 (S2c), authors should include, either in the text or in figure caption, their phrase regarding the highly warped Fermi surfaces (end of point 3 in their reply on page.12), this will further clarify their results.

We thank the reviewer for his/her important comment. Following the reviewer's suggestion, we have added the

relevant sentences in the end of Supplementary Note 2 as follows.

... The anomaly corresponding to the CDW transition is discerned ... in Supplementary Note 6. We note that the abrupt increase of resistivity on cooling below the transition temperature should be reflecting the disappearance of the Fermi surface at around \bar{M}_2 . On the other hand, there remain the Fermi surfaces at around \bar{M}_1 and also around $\bar{\Gamma}$ points. The carriers on these highly warped Fermi surfaces should be responsible for keeping the conductivity until low temperature.

b) authors may consider to slightly modify Figure 5 panel i). In fact, being panel i) extracted from h), and being the difference very small (as clearly seen in the corresponding figure in the supplementary), I would suggest to select e.g. only the 0.2 eV to -0.2 eV section to highlight the presence of a difference.

We thank for his/her important suggestion. Following the reviewer's suggestion, we have modified Fig. 5i as displayed in Fig. R1.

Fig. R1: The new version of Fig. 5i (Spin-resolved spectra near E_F at $\theta = \pm 4^\circ$).

We believe these revisions improve the quality of our present work for publication.

Reviewer #3 (Remarks to the Author):

I have reviewed the response by Mitsubishi et al. It is clear that the authors have invested a significant amount of effort to address my concerns, as well as those of the other reviewers. The authors have addressed or responded to my concerns regarding the presentation of their data, including the addition of spin-resolved results, the doping levels, and organization of the manuscript.

We sincerely thank again Reviewer #3 for his/her invaluable time and effort for reviewing our revised manuscript. We are glad to see that he/she acknowledges the improvement of our manuscript in the revised version.

However, the remaining concern of mine is the most important one, and is also echoed in the comments of Reviewer #2 ("...the presented interpretation is not conclusive in that sense...as the topology of a material is a property of the system as a whole, the data seems to demonstrate that the topology is conserved across the CDW transition...this is "the" focal point/question of the paper and needs to be addressed more convincingly").

Unfortunately, it appears that (despite the authors' efforts) the DFT calculations are not able to answer this question conclusively, presumably due to the complex crystal structure. The authors make some arguments which are plausible, but not concrete. Without a clear resolution of this issue, which (in my opinion) is the key result of this paper, I am still reluctant to recommend this manuscript for publication.

We thank the reviewer for his/her important comment. As the reviewer pointed out, we could not solidly construct the DFT-based topological arguments regarding the low-temperature $1T'$ phase. Nevertheless, we would again like to emphasize the valuable aspects of our present work: (i) the experimental findings of CDW-coupled band inversion and topological surface states, and (ii) the interpretation based on the local chemical bonding picture and crystal symmetry arguments. We believe that these insights should offer immense interests from researchers in the emergent field of topological quantum physics and materials, thereby promoting further sophisticated experimental / calculational investigations for CDW-coupled topological matters.